# Structural insights into scaffold-guided assembly of the Pseudomonas phage D3 capsid

Anna K. Belford, Joshua B. Maurer ⓘ, Robert L. Duda ⓘ, Alexis Huet ✉ &
James F. Conway ⓘ ✉

Tailed bacteriophages comprise the largest structural family of viruses with close relatives in archaea and the eukaryotic herpesviruses. The common assembly pathway produces an icosahedrally symmetric protein shell, called capsid, into which the double-stranded DNA genome is packaged. While capsid sizes and amino acid sequences vary considerably, the major capsid protein (MCP) folds are remarkably similar throughout the family. To investigate the mechanisms governing capsid size, we characterize the procapsid and mature capsid of phage D3, which expresses an icosahedral lattice with Triangulation number T = 9. We find that the MCP scaffold domain binds to the interior capsid surface, acting as a clamp to constrain subunit interactions. Following scaffold digestion, the MCP capsid domains form strong interactions that maintain capsid structure throughout maturation. The scaffold constraints appear critical for capsid size determination and provide important understanding of the factors governing capsid assembly in general and expands our understanding of these ecologically and biomedically important viruses.

The HK97 fold is shared by the major capsid proteins (MCPs) of all members of the large family of double-stranded DNA (dsDNA) tailed bacteriophages as well as the eukaryotic herpesviruses, some archaeal viruses, and bacterial compartments such as encapsulins and gene transfer agents[1–3]. The shapes and sizes of capsids assembled with this fold range from icosahedrally symmetric encapsulins with 60 copies of MCP arranged with Triangulation number T = 1[4] to the HK97 capsid itself with 415 copies of MCP and T = 7[5] up to the massive phage G with 3115 copies of MCP and T = 52[6], as well as prolate icosahedral capsids such as phage T4[7]. The remarkable flexibility of the HK97 fold to adopt this huge variety of structures is matched by its fidelity to produce the correct endpoint in each case. However, the mechanisms regulating capsid size and shape remain unknown.

In the case of the well-studied HK97 coliphage, assembly of the first complete shell, Prohead 1, is believed to be nucleated by the portal, a dodecameric ring around which hexameric and pentameric capsomers of the MCP bind with the aid of a 102-residue scaffold domain (also termed the Δ-domain) encoded as an N-terminal extension of the MCP (Fig. 1). After completion of Prohead 1, which in the case of HK97 contains 415 copies of the MCP, ~60 copies of the protease[8] and 12 copies of the portal, the scaffold is cleaved and released from the capsid to yield Prohead 2, a metastable particle primed for DNA packaging. Terminase encapsidates the phage DNA through the portal, triggering expansion of the capsid into the mature form, called Head, where covalent crosslinks form between MCP subunits[9,10]. A separately assembled tail binds to the portal of DNA-filled Heads to complete phage assembly. Unlike phages where the scaffold is a separately encoded protein, the HK97 domainal organization allows the prohead to be visualized before and after removal of the scaffold by controlling proteolysis, offering clues into how the scaffold influences capsid structure. Further, the presence of scaffold in a 1:1 ratio with MCP yields detailed structural determination of the scaffold-capsid interface, including around the symmetry mismatched portal vertex[11].

Department of Structural Biology, University of Pittsburgh School of Medicine, Pittsburgh, PA, USA. ✉e-mail: alh141@pitt.edu; james.conway@pitt.edu

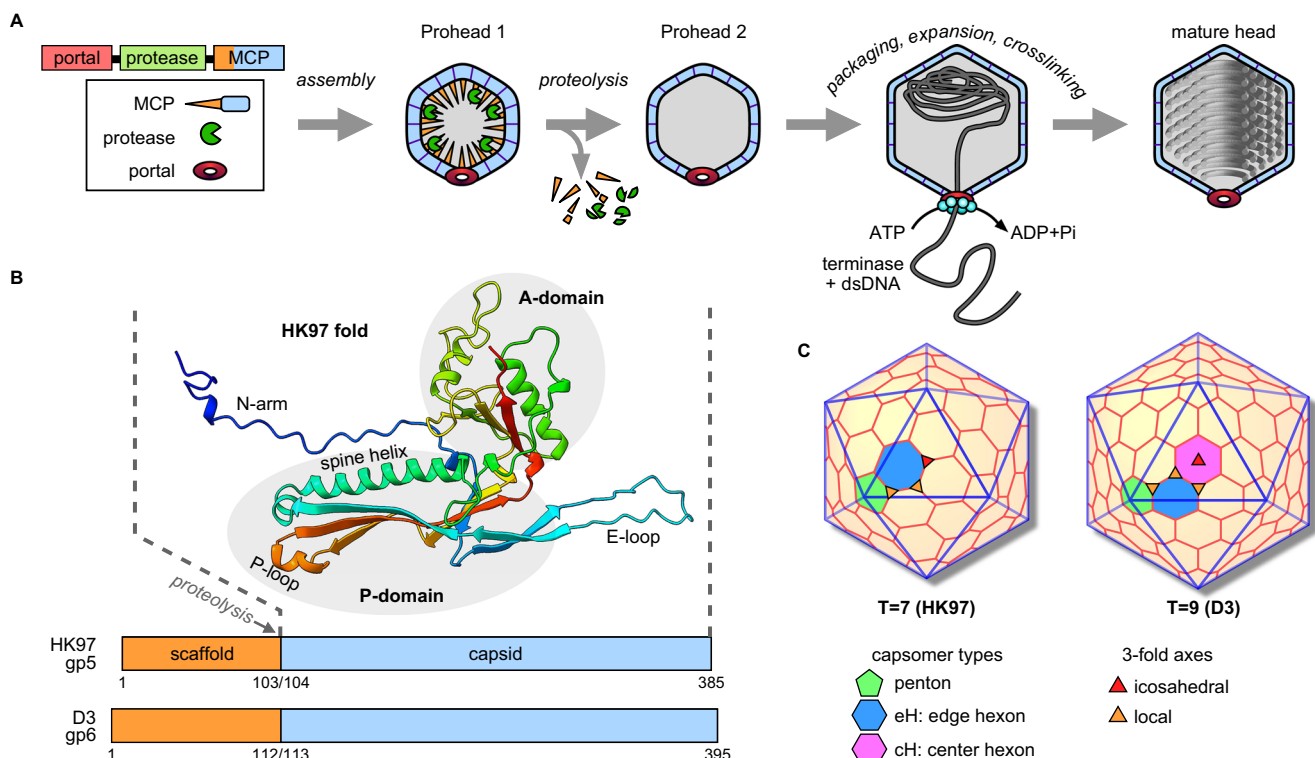

**Fig. 1 | Assembly and geometry of HK97-like capsids. A** Capsid assembly pathway of HK97-like capsids. The structural genes are sequential and in the order of portal (46 kDa), protease (32 kDa), and major capsid protein (MCP), which combines the scaffold and capsid functions (43 kDa). Expression yields the first complete shell, Prohead 1, that is subsequently proteolysed to form Prohead 2, a metastable particle primed for DNA packaging and expansion. The mature DNA-filled head binds the separately-assembled tail complex to form the infectious phage particle. **B** Fold of the capsid domain of phage HK97 MCP, including a comparison of the MCP domain lengths of phages HK97 and D3. **C** Comparison of icosahedral lattices for capsids expressing triangulation numbers 7 and 9. Images were generated by the Icosahedral Server of ViperDB[59]. The icosahedral threefold axis (red triangle) is located in the center of each triangular facet, while local threefold axes occur where three capsomers meet (orange triangles). An ASU of local threefold axes is marked for each triangulation number.

Here, we describe the capsid assembly pathway of *Pseudomonas aeruginosa* phage D3 and compare it with that of HK97. Both utilize an N-terminal scaffold domain and a capsid maturation protease, form covalent crosslinks during capsid maturation, and share 50% sequence identity in the capsid domain of the MCP. Despite these similarities, the D3 capsid expresses T = 9 geometry compared to the T = 7 of HK97, and we ask how the assembly pathway results in capsids with different sizes and the consequences on capsid maturation. We solved cryoEM structures for four assembly states of the phage D3 capsid, including the critical Prohead 1, where the capsid geometry is first expressed, through to the mature DNA-filled Head. Analysis of D3 proheads before and after the scaffold is removed reveals insight into the influence of the scaffold domain on capsid geometry. This understudied area of phage biology has direct implications for biomedicine, including therapeutic application of bacteriophages as alternatives to antibiotic molecules[12,13] and increased understanding of the structurally related herpesvirus capsids[14]. Further, the lytic relatives of temperate phage D3 are of particular interest for potential as antimicrobial agents against their host, *P. aeruginosa*, which the World Health Organization has designated as a high-priority pathogen due to its antibiotic resistance mechanisms[15].

## Results

### D3 capsid genes cloned in *Escherichia coli* produce Prohead 1 and Prohead 2

We produced D3 Prohead 1 and Prohead 2 particles by PCR-amplifying the D3 maturation protease and MCP genes from phage DNA and cloning them into the T7-expression vector previously used for HK97 capsid protein expression[16]. In subsequent steps, we added the portal gene and/or knocked out the protease by creating deletion or active site mutations. The resulting plasmids (Fig. 2A) all express the MCP gene and combinations of the portal gene and an active or inactive protease gene. Four plasmids were tested for Prohead 1 production—MCP with no protease or with an inactive protease, both with or without portal (plasmids #2, 3, 5, and 6). For Prohead 2, two plasmids were tested—MCP and wild-type protease with or without the portal (plasmids #1 and 4). All plasmids were expressed by autoinduction in *E. coli* and crude lysates from each were analyzed by native agarose gels (Fig. 2B) and by SDS PAGE (Fig. 2C). In agarose gels, particles the size of procapsids are sieved by the gel and form distinct bands, while hexameric and pentameric capsomers from dissociated proheads are not sieved and form a diffuse band. Plasmid #1 exhibited two distinct agarose gel bands (Fig. 2B): a sharper band that we assigned as Prohead 2 and confirmed by negative-stain electron microscopy (EM) of particles extracted from the band, and a more diffuse band consistent with capsomers. Thus, plasmid #1 produced both Prohead 2 and capsomers as opposed to plasmid #4 that shows one strong band where Prohead 2 is expected, suggesting that inclusion of the portal gene increased the yield of Prohead 2. The agarose gel patterns of the other plasmids (Fig. 2B, lanes 2, 3, 5, 6) are less clear, but all presented a sharper band that we ascribe to Prohead 1 and which is located just above the location of the corresponding Prohead 2 band on lanes 1 and 4. Similarly to the Prohead 2 gels, the Prohead 1 plasmid that included both the inactive protease and the portal gene (plasmid #6) exhibited the strongest band at the expected Prohead 1 location, indicating the highest yield. SDS-PAGE analysis of plasmids #1 and #4 (Fig. 2C) shows that the MCP is cleaved by the protease to the expected size (confirmed by N-terminal

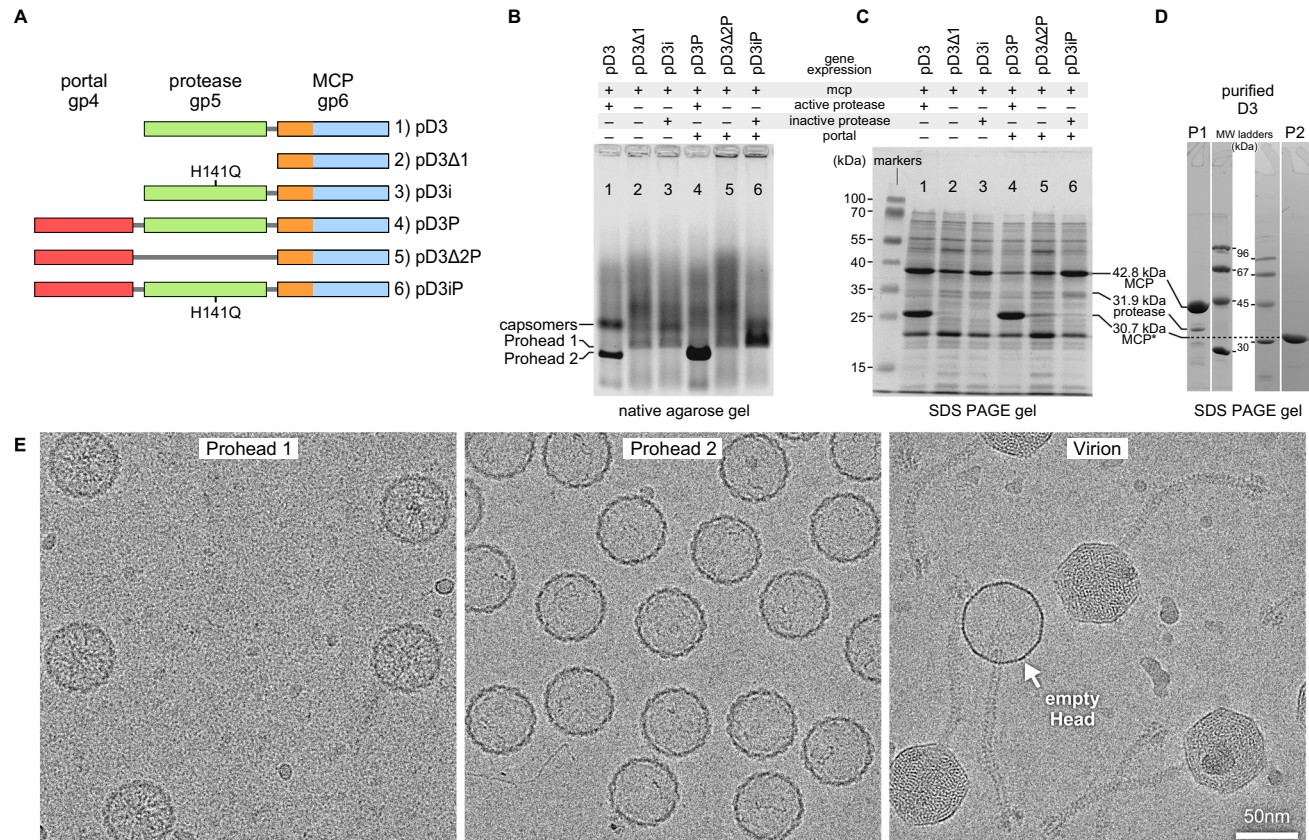

**Fig. 2 | Analyses of *P. aeruginosa* phage D3 prohead production in *E. coli* and the purified particles used for this study. A** The plasmids containing D3 capsid genes that were expressed using autoinduction methods[60] adapted originally for bacteriophage HK97 capsid proteins in 1.5 mL cultures (see "Methods"). An active-site mutation in the protease is indicated by H141Q. **B** Native agarose gel results for the plasmids in (**A**) with prominent bands for Proheads 1, 2 and capsomers labeled. Note that migration in agarose gels is dependent on size and surface charge, so migration distance may not correlate with size. **C** SDS-PAGE analysis corresponding to (**B**) with bands for full-length and proteolysed MCP indicated as well as the viral protease. **D** SDS-PAGE analysis of phage D3 particles purified from plasmids #6 (pD3iP−Prohead 1) and #4 (pD3P−Prohead 2). Note that the amount of protease was much less than the MCP amount: we calculated that there are ~90 copies of protease per Prohead 1 using band intensity measurements from the P1 gel. Gel experiments were repeated at least 3 times in each case. **E** Visualization of purified samples by cryo-electron microscopy. Proheads did not incorporate portals, while Virions included a population of empty particles termed "empty Head". Counts of exposures in each dataset are reported in Table 1 and Fig. S12.

sequencing) and that the yield of cleaved MCP is higher in plasmid #4 that includes the portal gene. For Prohead 1, plasmid #6 with the portal gene and the inactive protease had a stronger MCP band than plasmids #2, 3, and 5 (Fig. 2C). Although addition of DNA containing the portal gene increased the yield of D3 prohead particles, we could find no evidence that the portal protein was synthesized or incorporated into proheads, an unexpected result that will be explored in future work.

**D3 capsid morphologies**

D3 proheads made using plasmids #6 (pD3iP−Prohead 1) and #4 (pD3P −Prohead 2) were purified using methods similar to those used for HK97 proheads (see "Methods"). Purified Prohead 1 showed prominent bands for the uncleaved MCP and protease in SDS gels, while purified Prohead 2 contained only a cleaved MCP band (Fig. 2D). The purified Prohead 1 and Prohead 2 samples were vitrified and imaged by cryoEM for structure determination, in conjunction with a purified virion sample that included particles with and without packaged DNA (Fig. 2E). Structures for all four particle types were solved by imposing icosahedral symmetry on the entire capsid followed by vertex-focused extraction with fivefold symmetry imposed to increase the resolution[11] (see "Methods"). Final resolutions of the entire capsid maps were all ~3.5 Ångstroms (Å), while the vertex-focused maps reached 3.2 Å, 3.5 Å, 2.7 Å, and 3.5 Å for Prohead 1, Prohead 2, the DNA-packaged virion, and

the emptied virion, termed empty Head, respectively (Figs. S1–S3 and Table 1).

D3 procapsids and capsids were found to be larger than those of HK97 and the density maps revealed T = 9 geometry corresponding to 540 copies of the MCP arranged as 12 pentamers (in the absence of portal) and 80 hexamers (Fig. 3), compared to the HK97 T = 7 capsid with only 60 hexamers (Fig. 1C). Both D3 Proheads had the same vertex-to-vertex diameter of 590 Å, while the expanded capsids of both DNA-filled virion and empty Head had diameters of 750 Å. Prohead 1 had internal projections of strong density under each capsomer (Fig. 3A) that we attribute to the scaffold domains. These features were missing in Prohead 2 where the scaffold domain has been removed by proteolysis, as expected[5]. Notably, removal of the scaffold domains correlated with significant changes in the procapsid shell (Fig. 3A, B and Movie S1), illustrating the role of scaffold in overall capsid conformation as described below. The capsid walls of virions and empty Heads were thinner and more polyhedral, as is typical of mature capsids. Compared to the Proheads, all MCP subunits are displaced outward and capsomers adopt a more flattened and symmetrical conformation[17]. We also observed a small but distinct increase of shell diameter for empty Heads compared to DNA-filled virions that was most pronounced at the icosahedral faces (Fig. 3A), a counter-intuitive phenomenon that we have previously observed with the capsids of HK97 and T5[18,19].

**Table 1 | Microscopy and reconstruction details**

| | | Prohead 1 | Prohead 2 | Virion | Empty head |
|---|---|---|---|---|---|
| Microscope | | Krios 3Gi | Krios 3Gi | Krios 3Gi | Krios 3Gi |
| Magnification | | 75kx | 75kx | 96kx | 96kx |
| Camera | | Falcon 3 | Falcon 3 | Falcon 4i | Falcon 4i |
| Pixel size (Å) | | 1.08 | 1.08 | 0.82 | 0.82 |
| Exposures | | 3408 | 2972 | 43546 | 43546 |
| Particle counts | Capsid | 6960 | 20938 | 11496 | 1584 |
| | Vertex | 31011 | 235067 | 320746 | 15686 |
| Resolution (Å) | Capsid | 3.5 | 3.5 | 3.5 | 3.5 |
| | Vertex | 3.2 | 3.5 | 2.7 | 3.5 |
| EMDB ID | Capsid | 70800 | 70832 | 70878 | 70884 |
| | Vertex | 70831 | 70834 | 70879 | 70887 |
| PDB ID | | 9OSB | 9OTH | 9OUS | 9OUZ |
| Clashscore | | 0.44 | 0.27 | 0.1 | 0.16 |
| Favored rotamers (%) | | 90.7 | 96.15 | 97.36 | 96.43 |
| Ramachandran favored (%) | | 94.42 | 93.6 | 95.8 | 95.32 |
| Molprobity score | | 1.43 | 1.08 | 0.83 | 0.89 |
| Bad bonds | | 0 | 0 | 0 | 0 |
| Bad angles (%) | | 0.85 | 0.58 | 0.47 | 0.59 |
| CA geometry outliers (%) | | 0.59 | 0.81 | 0.89 | 0.97 |

The T = 9 geometry of the D3 capsid includes three kinds of capsomer: a penton and two different hexons termed "edge" (eH) and "central" (cH) (Fig. 1C). The eH interfaces with one penton, three other eHs and two cHs, while the cH occupies the center of the icosahedral facet coincident with the icosahedral threefold axis, and like the penton it is surrounded by eHs. Before capsid expansion, eHs adopt a pseudo-twofold symmetry (Fig. 3B) typical of the skewed conformation observed for the single hexon type in T = 7 proheads[5,20]. In contrast, the cHs that are not present in capsids with T = 7 symmetry exhibit threefold symmetry in Prohead 1 with an organization suggestive of three dimers, while after proteolysis the cHs in Prohead 2 exhibit pseudo-sixfold symmetry (Movie S1). In virions and empty Heads, both hexon types adopt the regular hexagonal shape that is typical for a mature capsid.

Our density maps exhibited sufficient side chain information to build reliable atomic models for all the capsid types except for a few highly mobile regions, including the tip of the E-loop, sections of the N-arm, and scaffold in the Proheads. We fit the D3 MCP sequence into the density maps using ModelAngelo[21] (Fig. S3) to provide an initial atomic model for each of the 9 MCP copies within the asymmetric unit (ASU), which includes one penton subunit, six from the eH, and two from the cH (Fig. 3B–D).

### Scaffold organization

The defining feature of Prohead 1 is the 113-residue scaffold domain of the MCP that is absent in the other capsid forms. The scaffold domain density appears as towers protruding into the capsid interior that connect to hairpins bound to the interior surface of each capsomer (Fig. 4A, B). The three capsomer types have markedly different protrusion architectures: under eHs, we observe an arch straddling the local twofold symmetry axis, while the density under the penton is organized as a short and stout tower that connects symmetrically to each capsid domain within the capsomer. The cH towers are ~90 Å-long cylinders of density that are stronger than for the other capsomer towers. As we found previously in HK97[11], the resolution of these protruding density towers is insufficient for atomic modeling, presumably due to flexibility, but we interpret them as being composed of

α-helical coiled-coil bundles comprising the N-terminal 70–80 residues of the scaffold domains that are predicted to adopt a ~ 90 Å-long coiled-coil conformation (Fig. S4B), which is consistent with the dimensions of the tower density (Fig. S4C). We propose that the penton and cH towers are formed by 5- and 6-stranded coiled-coils, respectively, consistent with the connected scaffold density, while the eH tower appears to be organized as two trimers, which reflects the local twofold symmetry of the eH capsomer (Fig. 4B).

In contrast, we could unambiguously solve the structure of the scaffold C-terminus between residues P81 and A113, which folds into a helix-turn-beta strand motif (Fig. 4C, D–orange) similar in size, though differing in sub-domain orientations, to the analogous region of the HK97 scaffold that we previously termed "spoke"[11] (Fig. S5). This region of the scaffold interacts with the end of the P-domain of the canonical HK97 fold, and we term it the P-domain Associated Region (PDAR). The PDAR binds to the mature MCP fold with its helix (residues 81 to 94) interacting with the end of the MCP spine helix and its beta strand (residues 104 to 109) extending the P-domain beta-sheet. These two secondary structure elements of the PDAR also interact with each other and bracket a U-shaped linker (residues 95 to 103) that orients them anti-parallel with a hydrophobic core at their interface (Fig. 4E). This hydrophobic core aligns with a hydrophobic patch on the P-domain of the MCP, further enhancing the interface between the PDAR and P-domain. Density corresponding to the N-terminal arm of the mature MCP, from residues 114 to 137, is not visible, leaving a 24-residue gap in our model. This flexibility of the first ~24 residues of the N-arm domain might expose the specific maturation cleavage site to the maturation protease.

PDARs from subunits of three different capsomers are organized around each local threefold location as trimers (Figs. 5 and S6A). However, while the PDAR trimers exhibit similar shapes and are quasi superimposable, they are not perfectly threefold symmetric (Fig. S6B). The trimers involving PDARs from hexons, ADC and BHE, include a similar density bridge between two of the subunits (Fig. 5) that correspond to the side chains of R102 (from subunits B and D) and R106 (from subunits E and A) which flank F364 (B/D) from a region of the P-domain called the P-loop (Fig. 5B, top and middle row; Fig. 1B). Other residues possibly involved in the formation and stabilization of this bridge are indicated in Fig. 5B. The third trimer, IFG, involving a PDAR from the penton and two from adjacent eHs, does not exhibit such a density bridge although we still observe asymmetry within the PDAR with one of the three interfaces of the trimer exhibiting increased bridging density and a closer monomer association than the other two locations (Figs. 5 and S6C). At all three locations, R102 and R106 side-chains of opposite PDARs extend into the intracapsomeric space with the P-loop F364 adjacent but not sandwiched between them as in the other two trimers (Fig. 5B, bottom row). These two residues are oriented relatively close to one another and may be stabilized by surrounding acidic and polar residues, including E353, E359, & E374 from the I subunit, and E359, D361, D363, F364, and E365 from the F subunit. This variability amongst the PDAR trimers likely reflects quasi-equivalence in the icosahedral asymmetric unit, but we note that the limited contacts within the PDAR trimers leaves a gap at the center of all the trimers where no interactions occur. Further, the PDARs within each trimer exhibit an electropositive pattern that suggests they do not form a tight cluster (Fig. S6D).

### The consequences of scaffold proteolysis

The role of the PDAR domain is further revealed by comparing the local threefold organization upon scaffold removal (Figs. 6 and S7). In Prohead 1 a gap in the capsid wall is surrounded by 3 arches of density originating from the adjacent P-loops that adopt an "up" conformation, while in Prohead 2 those arches are absent and the gap is filled by the P-loops now in a "down" conformation. The structural origin of this change is highlighted in Fig. 6B, emphasizing the spine helix (purple),

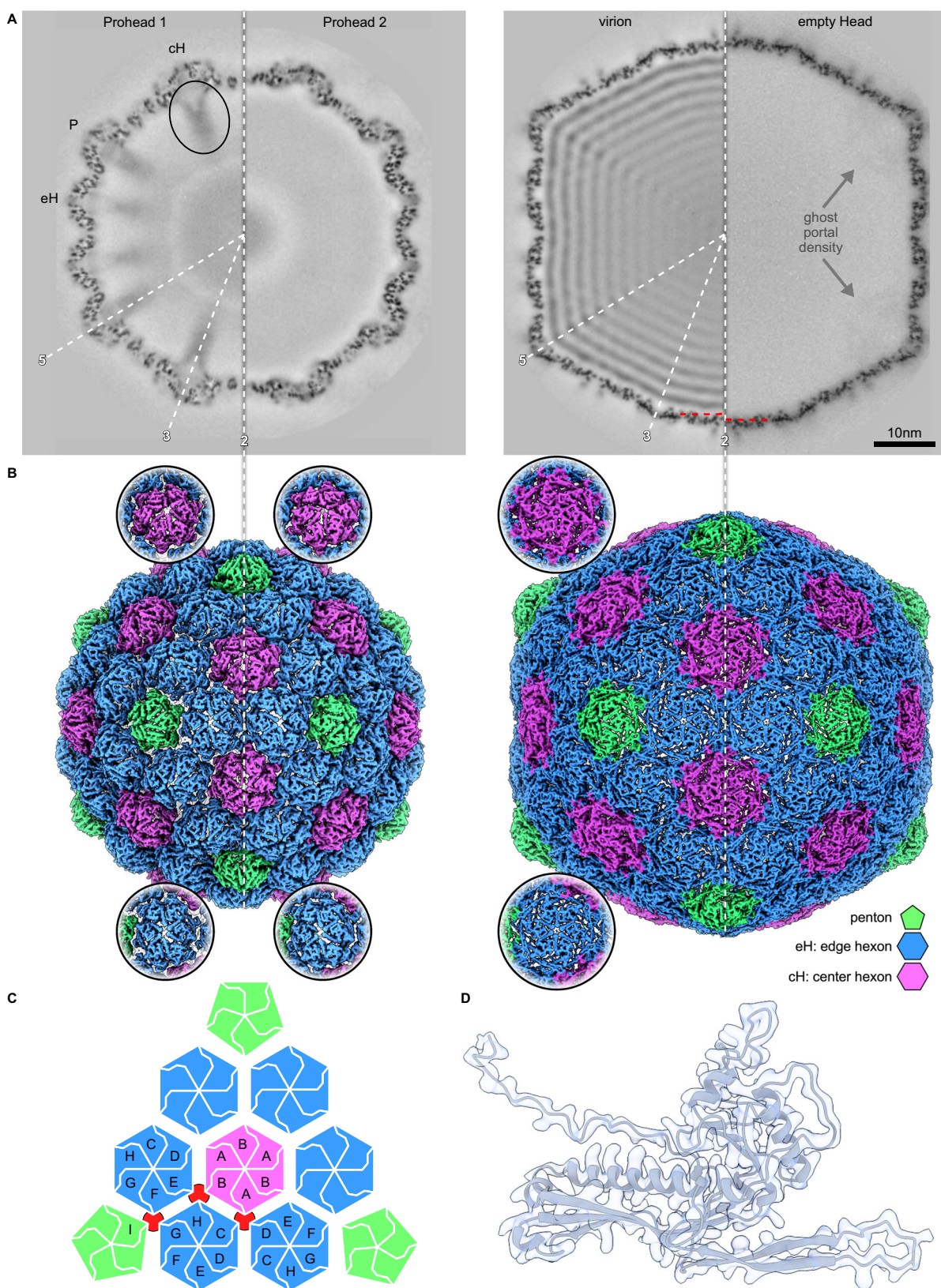

the P-domain (blue), and the PDAR (orange). The P-loop interacts with the PDAR in Prohead 1, but after its removal, these P-loops rotate and contact each other. Similarly, the spine helix of some subunits becomes straighter in Prohead 2, suggesting that the subunits are subjected to fewer constraints in absence of the scaffold. Superimposing the ASU subunits aligned by the core A-domain reveals an

overall higher conformational diversity in Prohead 1 than in Prohead 2 (Fig. 6C), while alignment of the Prohead 1 subunits by the PDAR shows an almost super-imposable conformation of the PDAR/P-domain complex (Fig. 6D). At the level of the capsid, Prohead 1 has a more angular conformation than the rounder Prohead 2 as measured by the dihedral angles between capsomers (Fig. S8). Variations in the angles

**Fig. 3 | CryoEM density maps of D3 capsid states. A** Central sections of the fours capsid states isolated, as indicated, with icosahedral symmetry axes indicated and the locations of specific capsomer types: P penton, eH edge hexon, and cH center hexon. The interior scaffold density under the Prohead 1 cH is indicated with an oval, while scaffold density is absent from all other capsid types. Red dashed lines indicate the difference in diameter at the twofold axis between the capsids of the DNA-filled virion and empty Head. Diffuse density beneath vertices of the empty Head ("ghost" density) represents the portal vertex averaged with 11 non-portal vertices—the absence of such density in Prohead 2 confirms that portals were not incorporated into the Proheads. **B** Surface views of the capsids with pentons colored green, eHs blue, and cHs purple. A schematic indicates the numbering of MCP subunits in the asymmetric unit and the specific interactions at the local threefold positions that are indicated in red. **C** Schematic representation of an icosahedral facet color-coded according to (**B**) and with the 9 subunit locations in the asymmetric unit marked. Positions of local threefold symmetry are indicated in red. **D** Density and model for a representative MCP subunit from the virion demonstrates side-chain density. See Fig. S3 for additional representatives of the four capsid states.

across Prohead 1 and the changes following proteolysis suggest that the scaffold might locally constraint the structure. Together, these observations indicate a local constraining function of the PDAR on the MCP capsid domains. We liken this constraint to a clamp that holds the capsid domain subunits in Prohead 1 to a network of limited interactions and which is released on proteolysis of the scaffold domains.

We note an additional conformational change that results from scaffold proteolysis. An isolated region of density contacts the P-domain in Prohead 2 at a site previously occupied by scaffold in Prohead 1 (Fig. S9). While this new density was insufficiently resolved to assign sidechains, its location adjacent to the C-terminus of the scaffold in Prohead 1 suggests that it corresponds to the MCP N-arm, as has been observed in HK97[11,22].

### Molecular basis of the morphological changes

The mechanism by which the PDAR maintains the P-loop in the "up" conformation is revealed in a close analysis of residues involved in the local threefold organization before and after the scaffold is removed (Fig. 7 and Movie S2). In Prohead 1, R102 of the PDAR is situated close to F364 of the same subunit, with sidechain distances ranging from 3.3 to 4.3 Å and possibly forming a cation-Pi interaction[23]. After scaffold removal abolishes any R102/F364 interaction, the F364 sidechain is now located within 3.3–4.6 Å from those of another arginine, R372, that was previously distant, as well as to N360, both of an adjacent subunit. This triad of sidechains is present three times at all local threefold locations, and results from reorganization of P-loops into the "down" conformation following removal of the PDAR R102 by scaffold proteolysis. The PDAR thus appears to prevent this trimeric interaction in Prohead 1, a restraint that may contribute to governing the geometry of Prohead 1 assembly.

Once the P-loop interface is established in Prohead 2, it is maintained during the subsequent expansion process to the mature Head (Fig. 8A and Movie S3), suggesting that it could act as a hinge holding the structure together during this massive conformational transformation. Although we have focused on the N360-F364-R372 interface, other residues are also involved in maintaining the structure during expansion, as shown in Movie S4. The final expanded structure also shows stabilization of the previously flexible E-loop and N-arm (Fig. 8B), enabling these structural features to be fully modeled. Both E-loop and N-arm interact together and are involved in numerous inter-capsomer interactions, forming a highly interconnected capsid structure, as described for HK97 and subsequently found in other phages. This complex network of interactions is mostly located around the threefold axes together with the newly made chemical cross-link between residues K178 and N367 (Fig. 8C), emphasizing the central role of this interface for capsid assembly, expansion, and stability[17,24–27].

### Discussion

Here we present the capsid assembly pathway of phage D3, showing the structural transition of an icosahedral capsid with T = 9 geometry from the first complete shell, Prohead 1, through proteolysis of the scaffold domain to Prohead 2, and expansion to the DNA-filled virion

and the empty Head. Assembly of capsids with the authentic size and shape requires only the full-length MCP that includes the N-terminal scaffold and C-terminal capsid domains, and as for phage HK97, the portal is not required. It should be noted that the HK97 assembly pathway was established without the portal and only recently has this special vertex been included in assembly analysis studies[11,28]. In particular, structures of capsids with or without the portal were virtually identical when icosahedral symmetry is imposed, demonstrating the relevance of portal-less capsids for studying assembly. By controlling expression of the viral protease, we observed two successive Prohead states—Prohead 1 that includes the scaffold, and Prohead 2 after scaffold digestion—while the 1:1 stoichiometry of scaffold with the capsid domain allowed the rigid PDAR subdomain to be resolved. Thus, the D3 MCP organization enables visualization of the early, geometry-defining Prohead 1 state, where the factors governing capsid assembly and shape can be investigated as well as the structural consequences of subsequent scaffold removal.

Studies of phage capsid assembly have shown that the MCP capsid domain has a significant role in specifying capsid size. For example, mutations in the MCP gene of phages P22 and λ produce small procapsids. In P22, the mutations are in the I-domain, an extra domain inserted into the P22 A-domain[29], whereas three small capsid mutants of λ map to a P-domain strand, the E-loop (near the tip) and the base of the T-loop of the P-domain[30]. Parasitic elements binding to the capsid exterior can also generate smaller capsids, such as for P2/P4[31], and ICP1/PLE[32]. Finally, mutants in the MCP capsid domain of HK97 show that there are interactions within and between capsomers with major roles in directing assembly of HK97 capsids to the correct size and shape[33,34]. Assembly is thus a cooperative process between the scaffold and capsid domains of the MCP.

Since capsid size and geometry are established early in assembly, we were particularly interested in comparing the initial prohead structures of phages D3 and HK97 that have closely related capsid proteins but assemble capsids of different sizes. Our results depict an assembly mechanism where the scaffold limits the molecular contacts between capsomers, allowing them to find the correct conformations for assembling a specific capsid geometry (Prohead 1) before establishing a more extensive network of interactions (Prohead 2) that is poised to transform into the expanded mature form with stronger bonds.

### The scaffold guides capsid geometry

Similarly to phage HK97, the scaffold domain of D3 can be divided into three subdomains (N-terminal coiled-coil region, a short linker, and the PDAR) that have distinct functions in regulating assembly. Although predicted to fold as a coiled-coil, the N-terminal region is not well resolved in Prohead 1 reconstructions, presumably due to flexibility, and appears only as towers of density that vary in overall organization according to location. Capsomer formation is likely to be promoted through bundling of coiled-coil subdomains from several subunits, but with flexibility to switch between penton, eH, and cH conformations (Fig. S10) as capsomers bind to the assembling capsid using a mechanism proposed for HK97 capsid assembly[34]. Such a

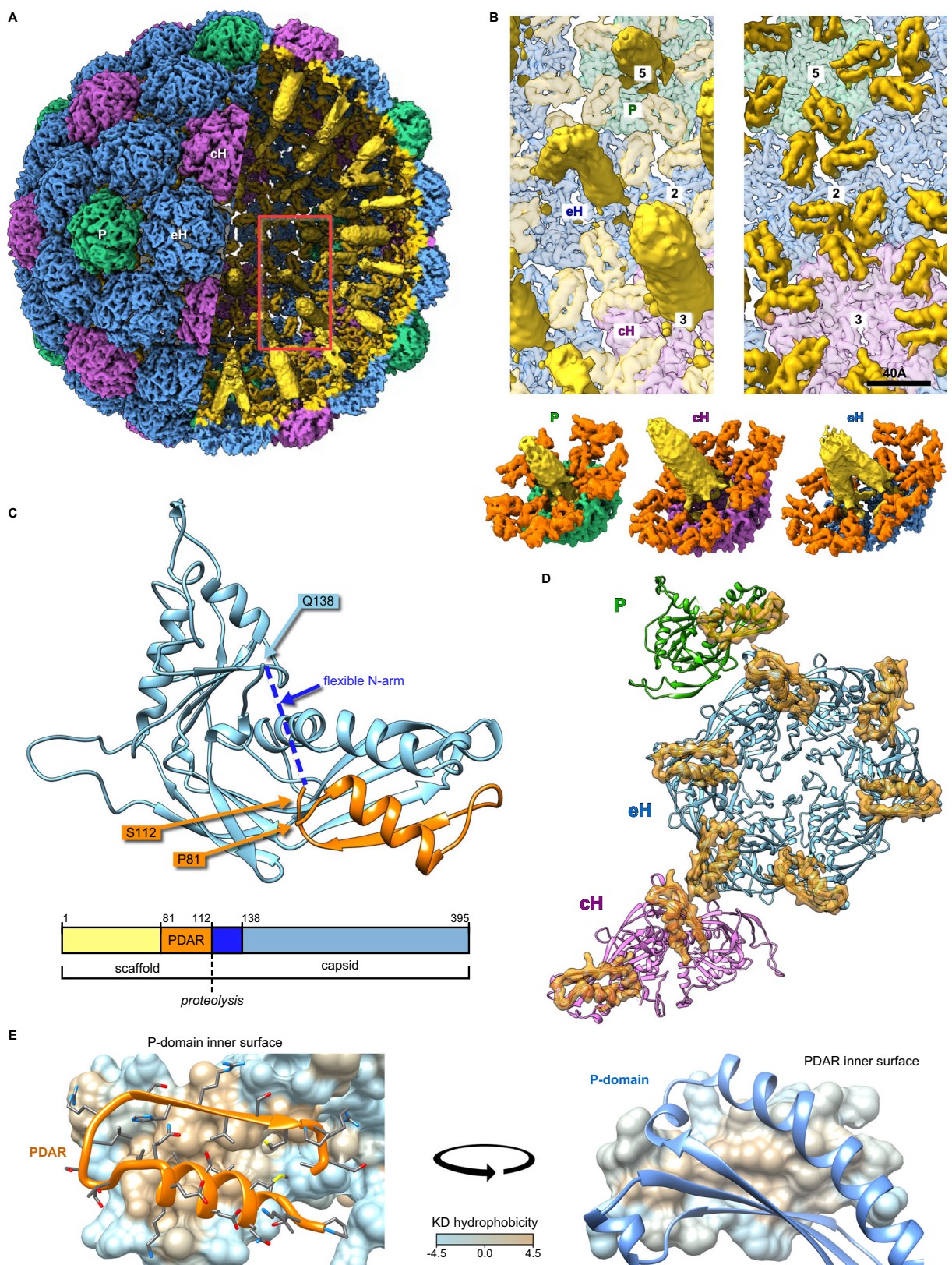

**Fig. 4 | Visualization and modelling of scaffold density. A** Towers of density protrude from the interior surface of Prohead 1 towards the procapsid center (yellow) corresponding to density indicated by the oval marker and similar density in the central section of Prohead 1 (Fig. 3A). The region indicated by the red box is enlarged in **B**, including a complete view of towers (yellow) under the penton, eH and cH, and a second view with the towers removed to show the remainder of the scaffold domain (orange) that binds to the interior surface of the procapsid. The bottom row shows excised capsomers and associated scaffold density, including the coiled-coil N-terminal towers (yellow), and the P-domain associated region (PDAR—orange). **C** A model of the D3 MCP derived from the density map, including the capsid domain (blue) and PDAR subdomain (orange). **D** A model of the ASU, including the capsid domains (purple, blue, green) and PDARs (orange). **E** Close-up views of a representative pair of a P-domain on the capsid interior surface and a PDAR, showing that the interface is hydrophobic as depicted with the Kyte-Doolittle hydrophobicity scale[61].

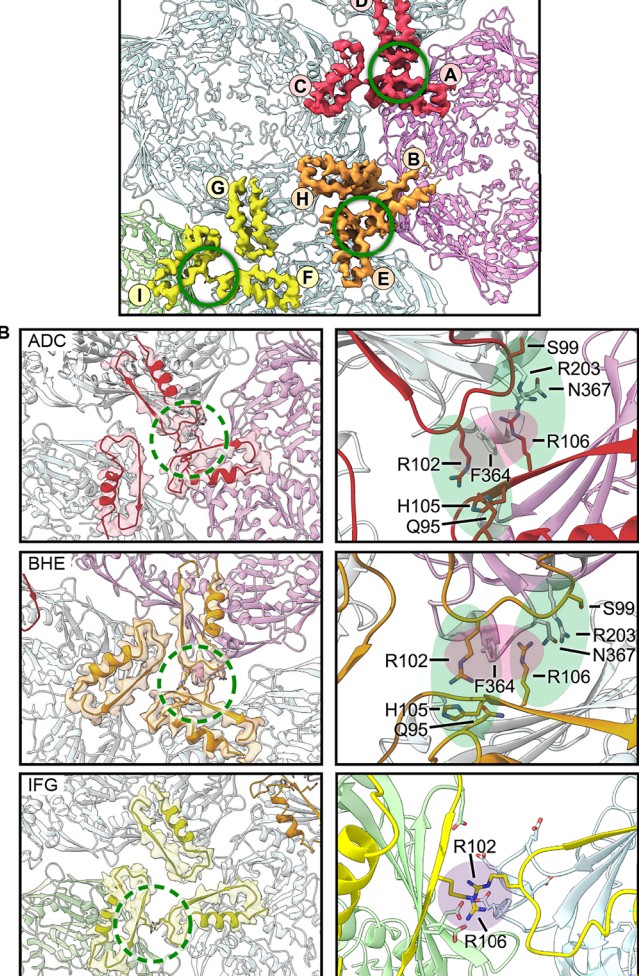

**Fig. 5 | PDAR interactions. A** Model and density maps of the three PDAR trimers at each local threefold location with the participating subunits labeled. Green circle highlights the PDAR-PDAR interface, which exhibits an increased network of interactions. **B** Left: Zoomed model and high transparency density maps of the three PDAR trimers at the intercapsomer interfaces. Right: Further zoomed model of each increased-interaction PDAR-PDAR interface. Colored ovals define the bounds of the likely-interacting residues.

conformational switch has been observed between pentameric and hexameric HK97 capsomers in solution[35]. Furthermore, capsid size correlates with the length of the coiled-coil subdomain, as exemplified by phages HK97 (T = 7), D3 (T = 9), and T5 (T = 13) (Fig. S4A). Such correspondence suggests that crowding of the scaffold domains in the capsid interior may preclude assembly of smaller capsid sizes while also serving to exclude cellular proteins from the interior[29,36]. However, such a size-by-exclusion mechanism is not sufficient to explain the geometry-specific accuracy of capsid assembly.

A strategy of imposing curvature on the assembling procapsid[37] is exemplified by the phage 80α system, where native capsids have T = 7 geometry but a parasitic genetic element called the *Staphylococcus aureus* pathogenicity island, SaPI1, expresses size redirection factors, Cmp, that modify capsid assembly to produce a smaller T = 4 shell[38]. The 80α scaffold and SaPI1 CmpB appear to compete for binding to the N-arm helix located along the spine helix of the 80α MCP, possibly changing the angle between capsomers, and thus capsid size. Similarly, the C-terminal PDAR subdomain of D3 is bound tightly to the P-domain of the HK97 fold and seems to act on the entire P-domain by bending the spine helix and imposing an "up" conformation on the P-loop such

that the P-domains of all subunits have a quasi-identical conformation, imposing local rigidity to the structure. This local rigidity manifests as a distinct angularity of Prohead 1 compared to Prohead 2 (Fig. S9) that is relaxed following removal of the scaffold. Such constraints suggest that the scaffold, and in particular the PDAR, aids in imposing specific dihedral angles between capsomers that effectively govern capsid geometry and size.

To play such a role, we propose that the PDARs from three adjacent scaffold domains, each associated with a different capsomer, form a clamp that restrains intercapsomeric interactions and enforces specific dihedral angles between the capsomers as they join the assembling procapsid. These clamps adopt a similar but not perfectly symmetric conformation at each local threefold site, as seen in Figs. 5A and S6B, and the geometry of the clamp is defined by small contact points between PDARs around residues R102 and R106, leaving an open triangle at the middle of the clamp. In comparison, the overall organization of PDARs is similar in the smaller HK97 Prohead 1[11], but the P-domains of HK97 form a tighter triangle where the subunits meet almost end-to-end (Fig. 9). We suggest that these differences may be a critical element that effects specific outcomes in capsid size as the network of PDAR interactions influences both the scaffold and capsid domains. For example, we observe strong density involving F364 flanked by R102 and R106 only around the central hexon in phage D3 (Fig. 5), suggesting an asymmetrical clamp-like activity of the PDAR throughout capsid. As capsomers assemble, such location-specific effects may influence the angle between capsomers and direct the assembly process towards a specific icosahedral geometry.

Despite the PDAR binding to, and imposing rigidity on Prohead 1, the entire scaffold domain is released completely from the shell by proteolysis[5]. We propose that the clamping action of PDARs is weakened by fragmentation of the scaffold, including the coiled-coil subdomains, and that transition of the shell to the rounder Prohead 2 conformation is both triggered by this fragmentation and acts to further dislodge the PDAR domains from the capsid interior.

## The dynamics of the local threefold sites

Digestion of the scaffold domain triggers conformational differences between Prohead 1 and Prohead 2 that are focused on the local threefold sites. A gap at this site in Prohead 1 becomes filled in Prohead 2 due to the three adjacent P-loops refolding from an "up" to a "down" conformation that increases intercapsomeric contacts. By enforcing the "up" conformation, the PDAR subdomains may allow the accreting capsomers to adjust to the binding environment of the assembling procapsid, associating either as a preformed capsomer or as capsomer subunits although whether the skewed hexameric capsomer conformation is induced before or during shell formation is unknown. Nonetheless, Prohead 1 is labile, and this is consistent with the low yield of Prohead 1 particles with a significant background of free capsomers following purification. In comparison, the conformational reorganization around the local threefold sites following removal of the scaffold yields the more stable Prohead 2, which has its P-loops refolded into the "down" conformation and does not dissociate. The formation of additional contacts is illustrated by the P-loop residue F364, which interacts with the PDAR of the same subunit in Prohead 1, but following scaffold removal, it contacts P-loop residues of adjacent subunits while moving toward the center of the local threefold and closing it. Increasing stable contacts at the local threefold sites may be a necessary step to prepare the procapsid for expansion without dissociating, as has been proposed for the HK97 capsid[22].

## Capsid expansion

Expansion results in further conformational changes at the local threefold sites, as previously observed in other dsDNA phage capsids

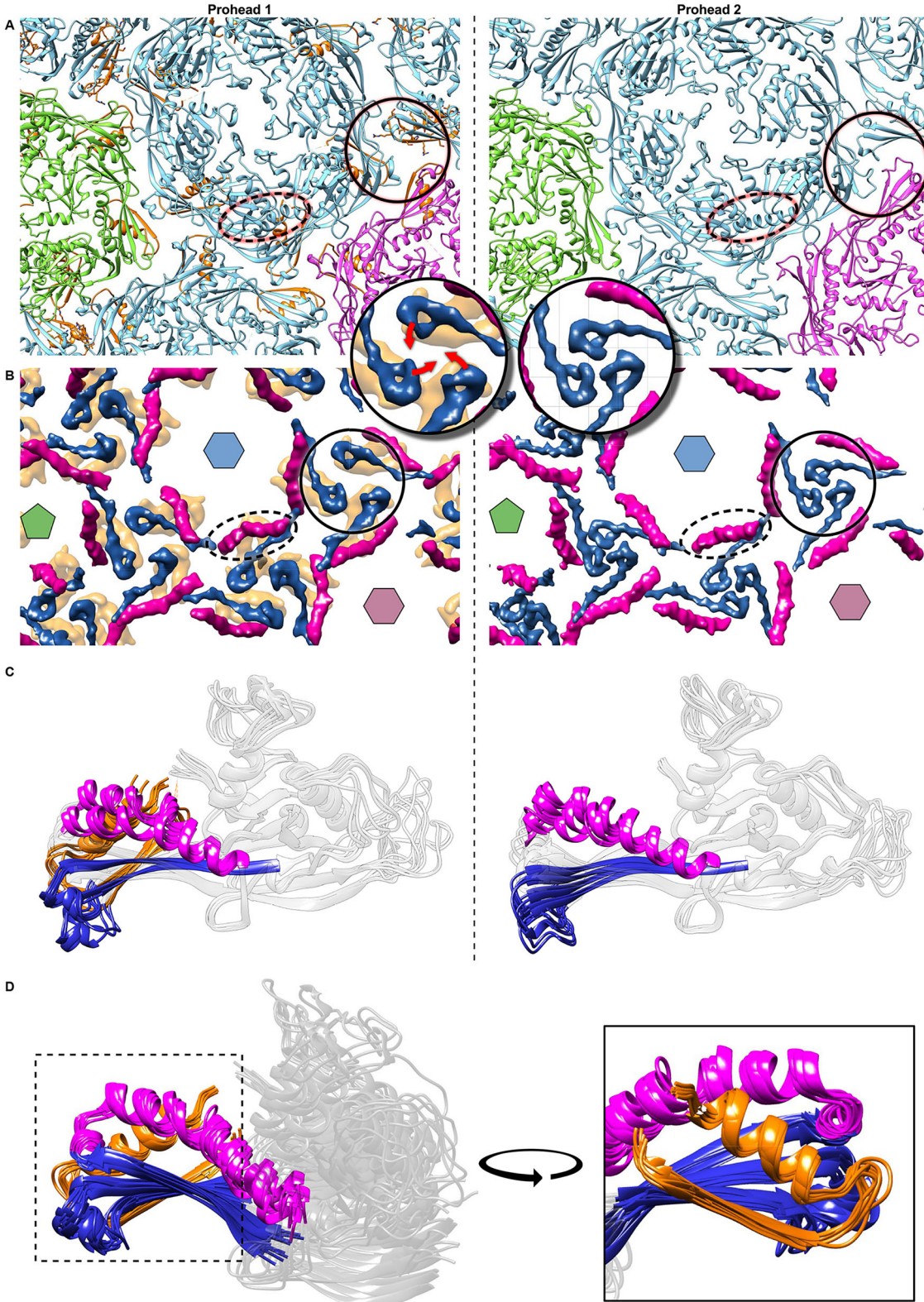

**Fig. 6 | Changes effected by proteolysis of the scaffold domain. A** Significant refolding occurs between Prohead 1 (left column) and Prohead 2 (right column) in the spine helix (oval dashed lines) and the P-loops adjacent to the local threefold axes−one axis is highlighted by a circle. Inset are twofold zooms of the circle, including arrows indication motion of the P-loops form the "up" to "down" orientations−also see Fig. S7. Capsomers are colored as in Fig. 1 (penton is green; eH is blue; cH is purple) and the scaffold PDARs are in orange. **B** A simplified model indicating the spine helices (purple), P-loops (dark blue), and PDARs (orange). **C** Alignment of 8 hexon subunits in the ASU by the A-domain on the MCP with regions colored as in (**B**) showing greater variability in Prohead 1 (left) than in Prohead 2 (right) as well as alignment into two sub-groups according to the spine helix overlaps in both cases. **D** An alternative alignment by the PDAR shows that the folds are remarkably constant throughout the ASU.

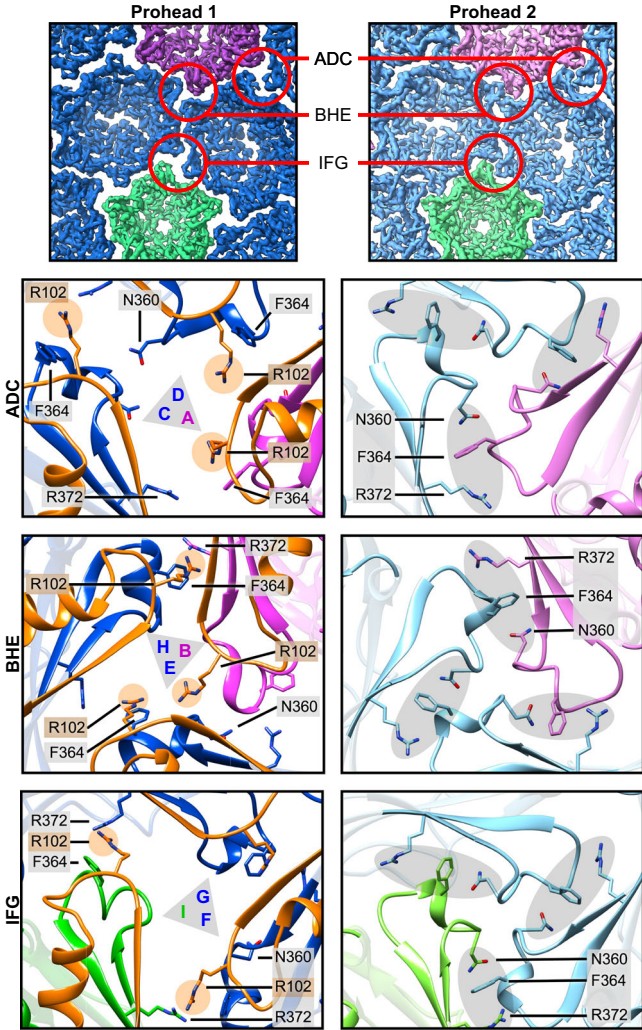

**Fig. 7 | Changes at the local threefold sites following proteolysis.** Sidechain interactions at the three different local threefold sites differ significantly before (Prohead 1) and after (Prohead 2) proteolysis. At each site, PDAR residues such as R102 appear to prevent a triad of capsid domain residues—N360, F364, R372—from making contact (at left) until after proteolysis (at right).

charged sidechains present in the interior of the D3 Head (Fig. S11B) could make contacts with the genome, perhaps drawing the capsid around a condensed dsDNA core[39]. A similar interaction was described in a set of simulation studies of HK97 DNA packaging, but no size change was calculated between the simulated empty and filled HK97 Heads[46].

### Generalization
We propose that the functions of the scaffold domains in directing assembly of phage D3 and HK97 capsids may be generalizable to dsDNA phages that encode the scaffold as a separate protein. While the resolvable density for scaffold proteins in cryoEM maps of phage proheads (other than for HK97 and D3) is generally weak, the strongest parts have usually been interpreted as the C-terminal (80alpha/SAP1[38], P22[41,47]) or the N-terminal end of the scaffold proteins (the prolate, partially expanded procapsid of phi29[48]) and found to contact an N-arm helix that lies against the backbone helix in the procapsid interior. The N-arms in each case form a helix that abuts the MCP backbone helix in a manner similar to the way the scaffold PDAR helices bind to the backbone helices in D3 and HK97, where one end of each of these scaffolds is connected indirectly to the MCP backbone helix. These direct (HK97, D3) or indirect (80alpha, SAP1, P22, phi29) connections of one end of the scaffold to the MCP backbone helix may be a common feature of all phage scaffolds and integral to their assembly-enabling function. Scaffolds are known to form oligomers in solution and are observed to do so within procapsids, where those oligomers appear to be dynamic—poorly resolved in our D3 and HK97 density maps but nearly invisible in other procapsid maps. Proposed functions of those oligomerization interactions include bringing MCP subunits together to form capsomers, bringing capsomers together, preventing or facilitating changes in capsomer structure, and influencing the dihedral angles between capsomers. Each of these functions would depend on a common linkage from the scaffold to the backbone helix of the MCP's P-domain, and additional studies on the dynamic regions of scaffolds will be necessary to understand the how they contribute to assembly.

### Limitations
This study resolves details of the phage D3 capsid and integrates the results with structural analysis of the similar but smaller HK97 capsid. While a common mechanism for regulating capsid assembly is emerging, structural analyses of related procapsids of the same size as well as with larger T-numbers are needed to validate and extend this model. The curious lack of portal gene expression, despite enhanced yield of Proheads 1 and 2 when the gene is included, remains to be understood. However, although portal may nucleate assembly in vivo, it does not seem to limit our study under conditions of over-expression.

## Methods
### Phage particle expression and purification
Dr. Andrew Kropinski, University of Guelph, Ontario, Canada, kindly supplied phage D3 for this project, and the host *P. aeruginosa* PAO1 was acquired from the Pseudomonas Genetic Stock Center. Details of primers, plasmids, bacterial strains, gene and protein sequences, and the biochemical methods are given in the Supplementary Information. Unprocessed scans of gels are included in the Source Data file.

### Negative stain electron microscopy
Purified samples of D3 Prohead 1 and 2 were diluted 20-fold in 50 mM Tris/50 mM Bis-Tris Propane HCl pH 6.0, 100 mM K Glutamate, and 20 mM Tris HCl pH7.5, 40 mM NaCl, respectively. Samples were applied to freshly glow-discharged graphitized carbon on copper grids, stained with 1% uranyl acetate, and imaged on a Tecnai TF20 microscope (Thermo Fisher Scientific—TFS, Waltham, MA, USA)

such as HK97[22], T5[39], λ[40], P22[41], and T7[42]. The flexible N-arm, which is not visible in the Prohead 1 density map and only weakly visible in Prohead 2, adopts a rigid conformation after expansion, binding to the E-loop and creating a complex network of intra- and inter-capsomer interactions. Following the HK97 paradigm, covalent crosslinks are formed between the exterior E-loops and the interior P-domains of subunits in neighboring capsomers around all local threefold sites that likely strengthen the D3 capsid in a manner similar to the threefold binding decoration proteins of non-crosslinking phages such as λ[43,44]. This crosslinking results in a topological interlinking of rings, forming a catenated protein that has been likened to chainmail[9,10]. The rings cross over each other around the local threefold axes, once more highlighting the importance of this site in capsid assembly. We observed that the diameter of the empty Head is slightly larger than that of the DNA-filled virion (Fig. S11A), which is unexpected as a powerful motor packages the genome of dsDNA phages to a concentration of ~500 mg/mL[45], and this encapsidated DNA would be expected to exert outward pressure on the capsid walls. However, similar size differences have been observed for other phages[18,19,42]. We suggest that the positively-

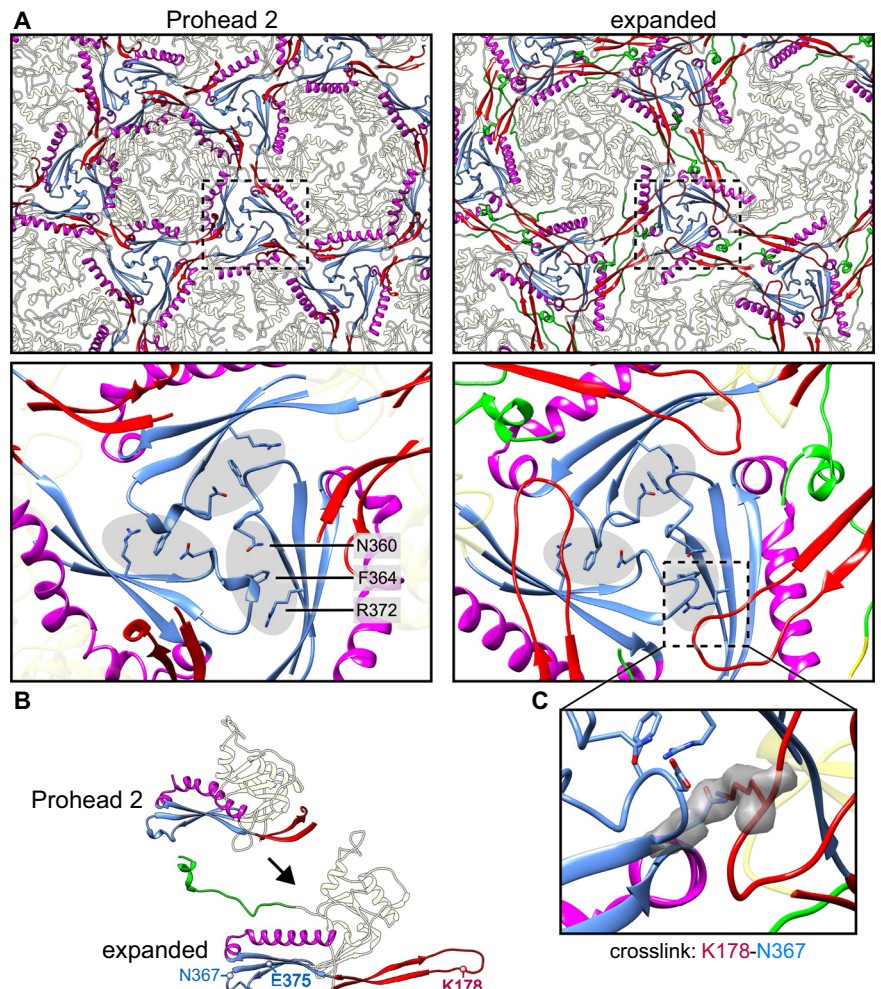

**Fig. 8 | Comparison of the capsid before and after expansion. A** Comparison of the local threefold interfaces between the Prohead 2 and expanded capsids (virion and empty Head) reveals a triad of sidechains—N360, F364, R372 (grey ovals)—that is maintained after expansion. **B** The N-arm (green) become stabilized. **C** The E-loop (red) crosslinks to the P-loop N367.

equipped with a TVIPS XF416 CMOS camera (TVIPS GmbH, Gilching, Germany).

## Cryo-electron microscopy

Three microliters of purified sample were pipetted onto a freshly glow-discharged Quantifoil R2/1 grid (Quantifoil Micro Tools GmbH, Großlöbicha, Germany) and then blotted and plunge-frozen with a Vitrobot Mk 4 (TFS). Grids were mounted on a TFS Krios 3Gi cryo-electron microscope operating at 300 kV and equipped with either a Falcon 3 (Proheads 1 and 2) or later Falcon 4i (virion and empty Head) direct electron detecting camera. Movies were collected in electron counting mode under the control of the TFS *EPU* software using a total dose of ~50 e/Å². Magnifications of 75,000 × or 96,000 × were used, correspond to 1.08 Å and 0.83 Å per pixel at the sample, respectively.

## Image processing

A cryo-EM data processing workflow is shown in Fig. S12. Phage particles were picked automatically with the e2boxer program of EMAN2[49] version 2.22 and SPHIRE-crYOLO[50] version 1.78. Reconstructions were carried out with RELION[51] versions 3 and 4, including motion correction[52], contrast transfer estimation[53], orientation determination, and particle polishing. Since the Procapsid samples did not incorporate portals, icosahedral symmetry was imposed throughout. As a final step to improve resolution, we compensated for defocus gradient

through the 600–700 Å-diameter particles by extracting individual vertices and calculating a final reconstruction with C5 symmetry imposed[11]. This essentially maintains the imposition of icosahedral symmetry as the step from icosahedral to C5 is balanced by the 12-fold increase in extracted particle count. The "gold-standard approach" in Relion[54] was used for resolution estimation (Fig. S1). Local resolution estimation was performed in Relion using Resmap[55] version 1.1.4 (Fig. S2). Data for each sample are listed in Table 1.

## Model building and refinement

Initial atomic models of the D3 asymmetric unit were generated with ModelAngelo[21] from the B-factor sharpened final map and the D3 MCP amino acid sequence. The scaffold domain sequence was included for modeling Prohead 1, but only the C-terminal part, Pro81 to Ser112, that contacts the capsid interior could be fit as the density corresponding to the remainder was not sufficiently resolved and is presumably flexible. The initial models were then iteratively refined with ISOLDE[56] version 1.8 in ChimeraX[57] version 1.8. The quality of the final models was assessed by MolProbity[58] version 4.5 as reported in Table 1.

## Reporting summary

Further information on research design is available in the Nature Portfolio Reporting Summary linked to this article.

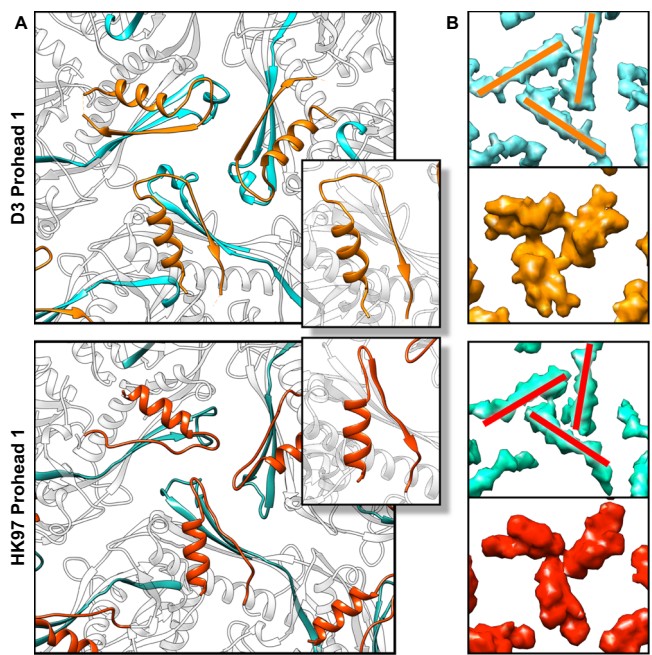

**Fig. 9 | Comparison of the PDAR/P-domain interfaces of phages D3 and HK97.**
**A** Model of a local threefold site where the PDAR is orange (D3) or red (HK97) adjacent to the P-loops (blue). Inset are PDAR models comparing the folds, including different loop dispositions linking the helix and beta-sheet. **B** The P-loops of HK97 form a tighter trimeric interface than for D3. **C** The PDARs associate with minimal overlap in HK97, compared to D3.

## Data availability

The source data underlying Fig. 2 and Supplementary Fig. S1 are provided as a Source Data file.

The final cryo-EM reconstructions are deposited with the Electron Microscopy Data Bank (EMDB) under accession codes EMD-70800 (D3 Prohead 1 capsid), EMD-70832 (D3 Prohead 2 capsid), EMD-70878 (D3 virion capsid), EMD-70884 (D3 empty head capsid), EMD-70831 (D3 Prohead 1 vertex), EMD-70834 (D3 Prohead 2 vertex), EMD-70879 (D3 virion vertex), and EMD-70887 (D3 empty head vertex), and the fitted coordinates with the Protein Data Bank (PDB) under accession codes 9OSB (D3 Prohead 1 vertex), 9OTH (D3 Prohead 2 vertex), 9OUS (D3 virion vertex), and 9OUZ (D3 empty head vertex). Further information and requests for resources and reagents should be directed to the lead contact, James F. Conway (james.conway@pitt.edu) and will be fulfilled under the terms of a Material Transfer Agreement. Source data are provided with this paper.

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

## Acknowledgements

The authors thank Katerina Toropova, Matthijn Vos, and Jeff Balkovec for their contributions to the early stages of this work. We thank Dr. Andrew Kropinski, University of Guelph, Ontario, Canada, for gifting D3 phage. This work was supported by National Institutes of Health grants R01GM047795 (R.L.D.) and R01GM144981 (J.F.C.). The Pittsburgh Center for CryoEM (RRID:SCR_025216) used for data collection in this project was supported, in part, by the University of Pittsburgh, the School of Medicine, the Department of Structural Biology, and the National Institutes of Health through grants S10OD019995 (J.F.C.) and S10OD025009 (J.F.C.). The content is solely the responsibility of the authors and does not necessarily represent the official views of the National Institutes of Health.

## Author contributions

R.L.D., A.H., and J.F.C. designed and supervised the project. R.L.D. made the initial clones of D3 capsid genes in expression plasmids and developed the purification protocols with refinements added by J.B.M. J.B.M. grew and purified D3 phage, created additional plasmids, characterized their expression, and did all purifications for the samples used for imaging. A.K.B., A.H., and J.F.C. collected cryoEM datasets and performed image reconstruction and molecular modeling. All authors were involved in interpretation of results and preparation of the manuscript.

## Competing interests

The authors declare no competing interests.
