## [Transparent Peer Review file · Nature Communications]

Structural insights into scaffold-guided assembly of the Pseudomonas phage D3 capsid

Corresponding Author: Professor James Conway

Version 0:

Reviewer comments:

Reviewer #1

(Remarks to the Author)

The study by Belford et. al., "Structural insights into scaffold-guided assembly of Pseudomonas phage D3 capsid" investigates the structural transitions and molecular mechanisms involved in the assembly and "maturation" of bacteriophage D3 capsids. Expression of the major capsid protein with or without protease (or inactivated protease) and the gp4 portal forming protein in E. coli. resulted in varying amounts of protein. Inclusion of the gp4 portal expressing vector generally increased capsid yield even though gp4 portal was not detected.

Cryo-EM was performed on three purified samples, Prohead-1, Prohead-2, and Virion resulting in 3.2 Å, 3.5 Å, and 2.7 Å (Virion) and 3.5 Å (Virion with Empty Head) maps respectively. Unlike HK97 which has T=7 icosahedral symmetry, D3 exhibited T=9 icosahedral symmetry, and was somewhat larger. The presence of the Prohead domain resulted in a premature like capsid conformational state revealed possible elements that govern conformational changes. A key structural element identified is the P-domain associated region (PDAR) which forms a variable trimeric complex at the 3-folds formed at interfaces between hexamers and pentamers of the capsid protein assembly. Atomic modeling revealed key rearrangements of this region which the authors hypothesize are critical to prohead maturation.

Generally, this manuscript was overly speculative. Most insights (some of which are very interesting and should be validated) were based solely on structure comparison and none were tested experimentally. Furthermore, there were an extensive number of items needing correction or clarification.

Major concerns

1) PDAR: This region is discussed extensively due to its novel determined structure and how it changes following maturation. There are multiple key residues identified (R102, R103, and F364 for example). Given the significant analysis of this region, and the hypothesis stated about it, some experimental validation is needed. Mutational analysis for example would make this observation and its impact on morphological changes less speculative.

"At all three locations R102 and R106 sidechains of opposite PDARs extend into the intracapsomeric space with the P-loop F364 adjacent but not sandwiched between them as in the other two trimers (Fig. 5B, bottom row)."

Provide images (zoomed in on this region) of the Map/Model so that the density supporting all sidechains discussed in all three flavors of the 3-fold interface are clearly visible to the reader.

2) Including the expression information as a primary figure (Fig 2) in the text is fine, but the expression is way more heavily interpreted than what is generally appropriate. If you insist on interpreting the gel as you have done, a western should be done to confirm the protein species. Also repeat the expression in triplicate and provide some form of densitometry analysis and graph with errors. Without this the last part of the first results section is based on qualitative analysis and not convincing. Maybe this section could be moved to the discussion section (although it is already too long).

Figure 2 is a protein expression comparison figure and does not rise to the level "Biochemical analysis" as titled in the figure legend.

3) "The defining feature of Prohead 1 is the 113-residue scaffold domain of the MCP that is absent in the other capsid forms. The scaffold domain density appears as towers protruding into the capsid interior that connect to hairpins bound to the interior surface of each capsomer..."

If the scaffold region cannot be modelled, and no attempt to do so is provided. The presence of this domain is one of the key novel observations. Were attempts made to resolve this region using alternative processing approaches. Focused alignments on this region with relaxation of C3 symmetry or multibody refinement could provide an improvement in map quality and detail. Even a slight improvement in the map here could allow for a much-needed inclusion of a model for this region.

"...but we interpret them as being composed of α -helical coiled-coil bundles comprising the Nterminal 70-80 residues of the scaffold domains that are predicted to adopt a $\sim 90^\circ$ -long coiled-coil conformation (Fig. S4)."

Providing a docked model in this map region would be more convincing.

"We propose that the penton and cH towers are formed by 5- and 6-stranded coiledcoils, respectively, consistent with the cH scaffold density, while the eH tower appears to be organized as two trimers, which reflects the strong local 2-fold symmetry of the eH capsomer (Fig. 4B)."

If you are going to propose this, build the model into the density and see if it "fits"

Intermediate concerns

1) Lack of consistency in quality that would lead to some confusion and frustration for a general audience. For example, there are multiple instances where figures are referenced out of order from how they appear. Panels from some figures are not referenced in the text. The term triad described an interaction between three things, but is used here to describe a series of three side chains where the first and third do not interact with each other.

2) I assume the below is "data not shown", if so, please provide a note to that effect.

"...confirmed by negative-stain electron microscopy (EM) of particles extracted from the band, and a diffuse band consistent with capsomers."

3) The term sharp band seems to be generous in the below text and qualitative. Suggest removing the text or providing additional evidence to allow for clear identification of the band.

"The agarose gel patterns of the other plasmids (Fig. 2B, lanes 2, 3, 5, 6) are less clear but all presented a sharp band that we ascribe to Prohead 1 and which is located just above the location of the corresponding Prohead 2 band on lanes 1 and 4."

4) Including a processing pipeline supplemental figure for each determined map would provide the reader with the information needed to assess the steps taken.

5) In the paragraph that begins with "D3 procapsids and capsids were found to be larger than those of..." The authors suggest that the removal of the scaffold is what causes the conversion of the prohead conformation to the head conformation. However, the data appears to suggest that the removal is a pre-requisite but is not what triggers the conversion. Instead, the data suggests that it is the packaging of the DNA that results in the change. Overall, this section and Fig 2 are very descriptive and doesn't provide significant insight.

6) Fig 5 and S6: The text provides a really nice description of the interactions, but some of these are not evident in the figures.

7) The authors appear to have interchanged the term P-loop and P-domain. Or P-loop is a region within the P-domain. Either way, P-loop is not defined in the text or figures.

8) "The structural origin of this change is highlighted in Fig. 6B, emphasizing the spine helix (purple), the P-domain (blue) and the PDAR (orange). The P-loop interacts with the PDAR in Prohead 1 but after its removal these P-loops rotate and contact..."

As depicted in Fig 6 and S7 the described structural changes cannot be assessed by the reader. An overlapping image would help.

9) "Variations in the angles across Prohead 1 and the changes following proteolysis suggest that the PDAR constraints at the local 3-fold interfaces propagate to the geometry of the shell structure, likely by playing a role in guiding capsid assembly. Together, these observations indicate a local constraining function of the PDAR on the MCP capsid domains. We liken this constraint to a clamp that holds the capsid domain subunits in Prohead 1 to a network of limited interactions and which is released on proteolysis of the scaffold domains."

The text beginning at the end of p8 is very speculative.

11) S2: Compare the same regions (the same alpha helix for example) of the four solved structures when demonstrating that the estimated resolution supports the modeling of sidechains.

12) A local resolution map is needed for all structures. Critical to this is clearly showing the PDAR region given the extensive analysis of this region.

Minor concerns/clarification needed

1) Why is there so little protease detected on both native agarose and SDS-PAGE gels in fig 2 B, C?

2) Label capsomer in Fig. 1 as well as define the differences between capsomer, virion, and head. For example, are the

virions in Fig. 2E considered capsomers?

3) Fig. 3: provide an overlap of the different hexons (e vs c) to highlight the differences.

4) Fig. 4: Panel E is not in the figure legend

5) Fig. 9: RMSD

6) Include the estimated sizes of each domain in panel Fig1A.

7) Fig1D—include a ladder that was run in parallel with the P1 and P2 fractions. Clearly delineate that the two samples are from cropped gels by separating them and/or placing a dotted line between them.

8) Does the presence or absence of packaged DNA occur randomly in a prep of Virions?

imaged by cryoEM for structure determination, in conjunction with a purified virion sample that included particles with and without packaged DNA (Fig. 2E).

9) Recommend not using "Head" as the description of the empty head of the virion. Empty-Head virion would be clearer to the naive reader.

10) Please provide a scale bar on the orthoslice images in Fig 2.

11) Provide the measured diameter of each and the observed change. "small but distinct increase of shell diameter"

12) What is meant by "most pronounced at the icosahedral faces"? Is the diameter the same at the 3-fold interface between head and virion? Again, scale bars and overlaid measurements would provide the reader with the needed information.

13) Figure 3: indicating where the symmetry axis described in the text are on the map would be helpful.

14) Figure 3C,D are not discussed in the text in a meaningful way and so are they needed?

15) PDBs and Maps were not available to download. (may be an issue with the portal).

16) Fig 7: The viewing angles chosen to depict the organization of R102 and F364 in prohead 1 does not clearly show the distance or relationship between the two amino acids clearly. This is not an issue for the viewing angles used for prohead 2.

17) A zoom in and distance measurement of the change of this described change would be good. "After scaffold removal abolishes any R102/F364 interaction, F364 is now adjacent to another arginine, R372,..."

18) Fig3: Indicate the final determined resolution for each map. The dotted line denoting the different maps shown in B does not line up with the seam of the pasted images.

19) Fig7: Clearly label the monomers for the alphabet three folds.

20) S6: the orientation of "CDA" in panel-A does not match "CDA" in panel-D

21) S3: were the measurements from the backbone or side chains, and if side chains are the modeled positions supported by density in the map.

22) S7: "Surfaces are colored by radius as shown in the scales" radius relative to what?

23) Fig. 6: Please clarify with a cartoon the difference between up and down conformations. Or place an arrow pointing which feature is up/down.

Reviewer #2

(Remarks to the Author)

This manuscript describes four cryo-EM structures capturing key assembly intermediates of the *Pseudomonas* phage D3 capsid: the initial Prohead 1 (with scaffold), the proteolyzed Prohead 2 (scaffold removed), the DNA-filled mature virion, and the empty head. The MCP's N-terminal scaffold domain serves as a "molecular clamp" on the interior capsid surface, whose constraint on subunit interactions in Prohead 1 is critical for determining the final T=9 capsid size. Subsequent proteolytic cleavage of the scaffold releases this "molecular clamp," triggering the formation of new, more stable inter-subunit contacts that prime the capsid for DNA packaging and expansion.

The authors provide high-quality structural data and a detailed description of the phage D3 assembly process. However, the work requires major revisions before it can be considered for publication. There are concerns regarding its data validation, method, and discussion. The major issues are as follows:

1. The structural biology research requires the provision of sufficient validation data, which is currently lacking in this manuscript.

Full PDB validation reports are not provided. While MolProbity scores are provided in Table 1, the full PDB validation reports are not included, making it impossible to assess the detailed quality of the atomic models in terms of bond lengths, bond angles, Ramachandran plots, and clashes. Full validation reports should be provided as supplementary material upon submission.

Local Resolution Maps are absent. The manuscript only reports the overall resolution, which can mask variations in quality across different regions of the structure. Without them, it is impossible to judge the quality of the density in key areas, such as the flexible E-loop or the functionally critical PDAR and P-loop interfaces.

2. The description in the Methods section is too brief and lacks critical details.

"Cryo-electron microscopy" Section: The description is too general. The authors should provide more detailed parameters, such as the specific defocus range used during data collection; the specific methods and software parameters for particle picking; the strategies and rounds for 2D and 3D classification; and the parameter settings for final reconstruction and refinement (Refinement/Polishing) in RELION.

"Structural analysis" Section: This description is also unclear. The authors mention using ModelAngelo and ISOLDE, but these are just tool names. The workflow for model building and refinement using ModelAngelo and ISOLDE needs to be described in detail, including how flexible regions were handled.

3. The authors mention performing local focused refinement on the five-fold vertices, which improved the resolution in these areas. However, the uniqueness of the T=9 lattice lies precisely in the two non-equivalent hexamers: the edge hexon (eH)

and the central hexon (cH). These hexons are located at or near the two-fold and three-fold axes. The authors do not mention whether they also performed local focused refinement on these equally critical regions. If optimization was only performed on the five-fold axes, it could lead to biases in the interpretation of the conformations and interactions of the hexameric regions (especially the much-discussed cH), and their model reliability would be lower than that of the five-fold regions. The authors should clarify their refinement strategy for the different symmetry axes and the final local structural quality.

4. The manuscript contains several instances of incomplete or erroneous literature citations.

For example, on page 6, "Before capsid expansion, eHs adopt a pseudo-2-fold symmetry (Fig. 3B) typical of the skewed conformation observed for the single hexon type in T=7 proheads 5,19": The citation here is problematic, being primarily outdated and not precise enough. The cited reference 19 is an early, low-resolution study. Subsequent work (e.g., studies on the capsids of phages HK97, P22, lambda) has revealed the details of the "skewed hexon" conformation at high resolution. The authors should update this citation to include more recent, high-resolution representative studies.

Another example on page 11: "Furthermore, capsid size correlates with the length of the coiled-coil subdomain, as exemplified by phages HK97 (T=7), D3 (T=9) and T5 (T=13)." This statement lacks any citation.

On page 12: "Expansion results in further conformational changes at the local 3-fold sites, as previously observed in other dsDNA phage capsids such as HK97, T5, lambda, P22 and T7." This sentence is also missing references.

On page 13: The citation 37 in "...strengthen the D3 capsid in a manner similar to the 3-fold binding decoration proteins of non-crosslinking phages such as lambda.37" is incorrect. A more appropriate reference matching the description would be (Structure. 2008, 16(9):1399-406 or Nat Struct Biol. 2000, 7(3):230-7).

5. The authors note that while inclusion of the portal gene enhanced particle yield, the portal protein itself was not detected in the purified procapsids. Under physiological conditions, the portal is the initiation point and nucleation center for capsid assembly (Sci Adv. 2023, 9(24):eadg8868; PLoS Biol. 2025, 23(4):e3003104). Its 12-fold symmetry and the inherent mismatch with the 5-fold axes of the capsid has effects on the entire geometric pathway of assembly. The T=9 capsids observed in this study are the products of spontaneous self-assembly via overexpression of high-concentration protein, without portal nucleation. Therefore, it cannot be ruled out that this in vitro self-assembly pathway and its final product are non-physiological artifacts of the experimental system. The authors should address this limitation in the discussion and cautiously argue the extent to which their observed structures represent the true in vivo assembly process.

6. Some structural descriptions appear to be based on uncertain observations. For example, the claim that an "isolated region of density" corresponds to the N-arm in Prohead 2 is difficult to assess, as it could not be modeled due to insufficient resolution (Figure S8).

The speculation that a "cation-pi interaction may form" between R102 and F364 in Prohead 1 is stated without providing the necessary data (distances) for validation. The authors should provide these details or show the fit of the side chains into the density.

7. In the discussion, the authors attempt to generalize their findings from D3 and HK97 (both with N-terminally fused scaffold proteins) to all dsDNA phages, including those with separately encoded scaffold proteins. There are differences in stoichiometry and conformational constraints between covalently linked and independently expressed scaffold proteins (e.g., covalent linkage ensures a 1:1 ratio and a fixed topological position). The authors should not simply equate the two but should discuss the differences between the systems in more detail and carefully define the scope of their conclusions.

Overall, the manuscript describes the structure of a T=9 phage capsid through its assembly pathway at near-atomic resolution. The work provides a solid structural basis for understanding how the scaffold domain acts as a "clamp" to guide capsid maturation, largely confirming the operation of known assembly mechanisms in a new system. This could be an interesting work, but considerable revision is required to address issues of data validation, methodological detail, and to more clearly define its specific contribution to the problem of capsid size determination.

Version 1:

Reviewer comments:

Reviewer #2

(Remarks to the Author)

The authors have addressed most of the concerns raised and implemented necessary revisions. I have no further comments.

Reviewer #3

(Remarks to the Author)

Review of revised manuscript by Belford et al. NCOMMS-

This manuscript describes the structures of the prohead and mature head of HK97-like Pseudomonas phage D3. The most interesting aspect of the work is their identification of the scaffolding protein and in particular the so-called PDAR (an acronym I am not fond of).

Reviewer 1 had a rather long list of comments and concerns. The main concern regarded the lack of non-structural

experimental validation of the authors' hypothesis of the involvement of the PDAR in capsid maturation, as well as numerous problems with the presentation and interpretation.

In my opinion, the authors have addressed the reviewer's concerns adequately in the revised manuscript and in their rebuttal. Specific concerns and the authors' response include:

1. The reviewer proposed experimental verification of the importance of residues R102 and R106 in the PDAR by mutational analysis. The authors' response was that this is outside the scope of the current structural paper. I tend to agree with the authors in this point.
2. "Provide images... of the map/model ... of the 3fold interface..." The authors have provided such images in Figure 5B, which show clearly the details of the 3fold interfaces.
3. Reviewer 1 had concerns about the interpretation of the protein expression gels in Fig 2. The authors explain that the figure is only meant to show the expression of the various proteins from the plasmid, not for quantitation of expression. I think the experiment shown is fine, since the bands on the gel are pretty unambiguous and the main point is the presence of the proteins in the reconstructions. Arguably, the proteins present in the D3 virions should have been identified by MS analysis, but this is not a major concern, since the focus in the paper is only on the major capsid protein.
4. "No attempt to model the scaffolding region was made." While modeling was not feasible, the authors have included an additional AlphaFold prediction of the scaffolding that matches the (unresolved) part of the towers seen in the map. Text has been added to explain the approach.

Other minor/intermediate concerns were addressed adequately. I have no particular issues with the revised manuscript.

REVIEWER COMMENTS

We appreciate the comments provided by both Reviewers and we give our detailed responses below. The points raised led to improvements in clarity and accuracy and we believe the manuscript has been strengthened as a result. Our responses are in red, with changes to the manuscript text in green.

Reviewer #1 (Remarks to the Author):

The study by Belford et. al., “Structural insights into scaffold-guided assembly of Pseudomonas phage D3 capsid” investigates the structural transitions and molecular mechanisms involved in the assembly and “maturation” of bacteriophage D3 capsids. Expression of the major capsid protein with or without protease (or inactivated protease) and the gp4 portal forming protein in E. coli. resulted in varying amounts of protein. Inclusion of the gp4 portal expressing vector generally increased capsid yield even though gp4 portal was not detected.

Cryo-EM was performed on three purified samples, Prohead-1, Prohead-2, and Virion resulting in 3.2 Å, 3.5 Å, and 2.7 Å (Virion) and 3.5 Å (Virion with Empty Head) maps respectively. Unlike HK97 which has T=7 icosahedral symmetry, D3 exhibited T=9 icosahedral symmetry, and was somewhat larger. The presence of the Prohead domain resulted in a pre-mature like capsid conformational state revealed possible elements that govern conformational changes. A key structural element identified is the P-domain associated region (PDAR) which forms a variable trimeric complex at the 3-folds formed at interfaces between hexamers and pentamers of the capsid protein assembly. Atomic modeling revealed key rearrangements of this region which the authors hypothesize are critical to prohead maturation. Generally, this manuscript was overly speculative. Most insights (some of which are very interesting and should be validated) were based solely on structure comparison and none were tested experimentally. Furthermore, there were an extensive number of items needing correction or clarification.

Major concerns

1) PDAR: This region is discussed extensively due to its novel determined structure and how it changes following maturation. There are multiple key residues identified (R102, R103, and F364 for example). Given the significant analysis of this region, and the hypothesis stated about it, some experimental validation is needed. Mutational analysis for example would make this observation and its impact on morphological changes less speculative.

We appreciate the Reviewer emphasizing the importance of the PDAR as presented in our report. However, the Reviewer seems to misunderstand – the PDAR is present in Prohead 1 only, and it is absent once the scaffold domain (of which the PDAR is a part) is proteolyzed. Thus, the structure of the PDAR doesn't change during maturation, its only conformation is that visualized in Prohead 1. To further emphasize that the coiled-coil and PDAR subdomains are both removed by scaffold proteolysis, we have indicated the point of proteolysis on the MCP schematics in Figure 1B and Figure 4C.

The second point raised is validating the significance of several key PDAR residues – R102 and R106 (not R103, apparently mistyped) – and their proximity to the P-loop domain residue F364, which is refolded after proteolysis (Prohead 2) since the PDAR is now absent. We are planning to test the effects of mutations in these residues, but such work is beyond the scope of our current manuscript, and the phenotypes of such mutations may in any case not be revealing. Indeed, our comparison of four capsid structures at high resolution as well as with the smaller capsids of the structurally related phage HK97 already provides a wealth of information about capsid assembly and maturation.

“At all three locations R102 and R106 sidechains of opposite PDARs extend into the intracapsomeric space with the P-loop F364 adjacent but not sandwiched between them as in the other two trimers (Fig. 5B, bottom row).”

Provide images (zoomed in on this region) of the Map/Model so that the density supporting all sidechains discussed in all three flavors of the 3-fold interface are clearly visible to the reader.

The left-hand column of Figure 5B already includes the density/model images requested by the Reviewer, and the sidechain density of R102, R106 and F364 is included. The right-hand column shows the modelled sidechains in higher detail. We believe that the concepts described in the text and illustrated in this Figure are already sufficiently apparent and don't need further emphasis.

2) Including the expression information as a primary figure (Fig 2) in the text is fine, but the expression is way more heavily interpreted than what is generally appropriate. If you insist on interpreting the gel as you have done, a western should be done to confirm the protein species. Also repeat the expression in triplicate and provide some form of densitometry analysis and graph with errors. Without this the last part of the first results section is based on qualitative analysis and not convincing. Maybe this section could be moved to the discussion section (although it is already too long). Figure 2 is a protein expression comparison figure and does not rise to the level "Biochemical analysis" as titled in the figure legend.

Figure 2 is not intended to provide a quantitative measure of expression levels but instead shows that the inclusion of the portal gene in the construction boosts expression, even though portals are absent in the purified procapsids. The mechanism by which inclusion of the portal gene influences assembly is unknown and a subject for on-going studies. Western blots were not done since there are no antibodies against the phage D3 structural proteins – instead we have confirmation of the major capsid protein through the cryoEM density maps. Absence of portal in the Proheads is verified by the cryoEM reconstructions, which do not exhibit any “ghost” of portal density in their cross-sections – compare, for example, the density under the 5-fold capsomers in cross-sections of Prohead 2 and the empty Head in Figure 3A. We updated Figure 3A to indicate this “ghost” density in the empty Head map and added a sentence to the caption that emphasizes this aspect:

Diffuse density beneath vertices of the empty Head ("ghost" density) represents the portal vertex averaged with 11 non-portal vertices – the absence of such density in Prohead 2 confirms that portals were not incorporated into the Proheads.

We changed the title of Figure 2 to: *Analyses of P. aeruginosa phage D3 prohead production in E. coli and the purified particles used for this study.*

3) "The defining feature of Prohead 1 is the 113-residue scaffold domain of the MCP that is absent in the other capsid forms. The scaffold domain density appears as towers protruding into the capsid interior that connect to hairpins bound to the interior surface of each capsomer..."
If the scaffold region cannot be modelled, and no attempt to do so is provided. The presence of this domain is one of the key novel observations. Were attempts made to resolve this region using alternative processing approaches. Focused alignments on this region with relaxation of C3 symmetry or multibody refinement could provide an improvement in map quality and detail. Even a slight improvement in the map here could allow for a much-needed inclusion of a model for this region.

The Reviewer is not correct in claiming that we make no attempt to model the scaffold. The complete second sentence quoted ends: "...of each capsomer (Fig. 4A,B)." These panels of Figure 4 show the subdomains of the scaffold colored in yellow (coiled-coil) and orange (PDAR). Our wording, as quoted above, indicates protruding towers as the most visible manifestation of the scaffold since the PDAR subdomains are hidden between the towers and the interior surface of the Prohead 1 capsid. We do, of course, model the PDAR subdomain, as shown in panel C, and this part of the scaffold is an important aspect of our work, as the Reviewer notes.

However, we did not include a model of the N-terminal coiled-coil subdomain density as the quality of the density is poor, which we attribute to flexibility due to the short linker connecting it to the rest of the scaffold/MCP and to the very different organization of this subdomain in non-equivalent positions, compared to the relatively identical PDAR organization in corresponding positions. Our previous visualization of scaffold coiled-coil subdomains in the Prohead 1 of phage HK97 (Huet et al, 2023) was successful only where the coiled-coil subdomains contacted the portal directly, but the D3 Prohead 1 sample does not incorporate portals. We had attempted a focused alignment and relaxed the symmetry, but this did not improve the quality of the coiled-coil subdomain density, confirming its flexibility. While this negative result was not included in our manuscript, we have now added panel C to Supplemental Figure S4 to show that an AlphaFold2 model based on the coiled-coil subdomain sequence is compatible with the dimensions of the density towers.

“...but we interpret them as being composed of α -helical coiled-coil bundles comprising the N-terminal 70-80 residues of the scaffold domains that are predicted to adopt a ~ 90 -Å-long coiled-coil conformation (Fig. S4).”

Providing a docked model in this map region would be more convincing.

“We propose that the penton and cH towers are formed by 5- and 6-stranded coiledcoils, respectively, consistent with the cH scaffold density, while the eH tower appears to be organized as two trimers, which reflects the strong local 2-fold symmetry of the eH capsomer (Fig. 4B).”

If you are going to propose this, build the model into the density and see if it “fits”

As in our response above, we have added a new panel C to Supplemental Figure S4 showing the fit of a coiled-coil bundle corresponding to the scaffold coiled-coil subdomains into the tower density under the Prohead 1 center hexon that further corroborates the assignment of the scaffold coiled-coil subdomains to this density. We feel the assignment of the coiled coil subdomains to these towers is more than adequately supported by this modeling and by the clear linker density connecting the towers to the N-termini of the adjacent PDAR density, as seen in Figure 4B (cited in our text, quoted above). We have modified the sentences quoted above as follows:

As we found previously in HK97 [11], the resolution of these protruding density towers is insufficient for atomic modeling, presumably due to flexibility, but we interpret them as being composed of α -helical coiled-coil bundles comprising the N-terminal 70-80 residues of the scaffold domains that are predicted to adopt a ~ 90 Å-long coiled-coil conformation (Fig. S4B) which is consistent with the dimensions of the tower density (Fig. S4C). We propose that the penton and cH towers are formed by 5- and 6-stranded coiled-coils, respectively, consistent with the connected scaffold density, while the eH tower appears to be organized as two trimers, which reflects the local 2-fold symmetry of the eH capsomer (Fig. 4B).

Intermediate concerns

1) Lack of consistency in quality that would lead to some confusion and frustration for a general audience. For example, there are multiple instances where figures are referenced out of order from how they appear. Panels from some figures are not referenced in the text. The term triad described an interaction between three things, but is used here to describe a series of three side chains where the first and third do not interact with each other.

We have checked the manuscript carefully: all Figures were cited in order, except Supplementary Figures S8 and S9 due to late editing of the manuscript, and which we have now corrected. Further, all Figure panels are referenced in the text except for several cases where the whole Figure is cited – we don’t believe that all panels need explicit citations where the caption provides the commentary.

Regarding our use of “triad”, we fail to understand the Reviewer’s point. We use the term only three times (fittingly) – once in the Results section of the text, and once in each of the captions to Figures 7 and 8 – and in each case we refer to the three sidechains F364, R372 and N360 in Prohead 2, the virion capsid, and the empty Head. In the Results section we write:

This new triad of sidechains is present three times at all local 3-fold locations, and results from reorganization of P-loops into the "down" conformation following removal of the PDAR R102 by scaffold proteolysis.

Whereas in the caption to Figure 7 we write:

At each site, PDAR residues such as R102 appear to prevent a triad of capsid domain residues – N360, F364, R372 – from making contact (at left) until after proteolysis (at right).

And in the caption to Figure 8 we write:

...reveals a triad of sidechains – N360, F364, R372...

We point out that the close spatial organization and likely interaction is retained from Prohead 2 onwards but is prevented from alignment in Prohead 1 due to sidechains from the scaffold's PDAR subdomain. Since the F364 sidechain is sandwiched between those of N360 and R372, as indicated in the text and Figures 7 and 8, we do not expect a direct interaction between N360 and R372, but nonetheless the three-sidechain grouping (triad) seems significant as it is established after proteolysis and maintained during capsid expansion.

2) I assume the below is "data not shown", if so, please provide a note to that effect.

"...confirmed by negative-stain electron microscopy (EM) of particles extracted from the band, and a diffuse band consistent with capsomers."

The complete sentence is: "Plasmid #1 exhibited two distinct agarose gel bands (Fig. 2B): a sharp band that we assigned as Prohead 2 and confirmed by negative-stain electron microscopy (EM) of particles extracted from the band, and a diffuse band consistent with capsomers." While we initially used negative stain EM to confirm the nature of the bands, as stated, the ultimate confirmation came from the cryoEM data collection and analyses that followed. Thus, we do not think it worth including negative stain electron micrographs as supplemental data, and journals generally disallow the use of "data not shown".

3) The term sharp band seems to be generous in the below text and qualitative. Suggest removing the text or providing additional evidence to allow for clear identification of the band.

"The agarose gel patterns of the other plasmids (Fig. 2B, lanes 2, 3, 5, 6) are less clear but all presented a sharp band that we ascribe to Prohead 1 and which is located just above the location of the corresponding Prohead 2 band on lanes 1 and 4."

Our experience over decades of analyzing phage capsid assembly intermediates by native agarose gel is that small particles (such as capsomers) produce more diffuse bands while larger particles (capsids) form distinct sharper bands and heterogeneous particles form smears. In these gels, migration depends on both the charge and the size of the particles. We added a sentence of explanation above the quoted sentence: *In agarose gels, particles the size of procapsids are sieved by the gel and form distinct bands, while hexameric and pentameric capsomers from dissociated proheads are not sieved and form a diffuse band.* and now refer to the bands as "sharper" and "more diffuse". As above, the ultimate confirmation of the nature of assembled capsid samples was provided by cryoEM. We also added this sentence to the caption of Figure 2:

Note that migration in agarose gels is dependent on size and surface charge, so migration distance may not correlate with size.

4) Including a processing pipeline supplemental figure for each determined map would provide the reader with the information needed to assess the steps taken.

Reviewer #2 also asks for more detail on the cryoEM analysis, and we have expanded on our text in Methods:

As a final step to improve resolution, we compensated for defocus gradient through the 600-700 Å-diameter particles by extracting individual vertices and calculating a final reconstruction with C5

symmetry imposed, as done previously [11]. Note that the signal is maintained by averaging the 12 vertices from each capsid and applying C5 symmetry.

However, while we appreciate the utility of a pipeline figure for following cryoEM analyses in general, the work in this report was very straightforward: icosahedral symmetry was imposed initially followed by a vertex-focused refinement. We did not perform further classification to locate the portal vertex since it is absent in both the Prohead 1 and 2 particles. We suggest that including a pipeline figure would serve little purpose beyond the updated Methods text and the data already in Table 1.

5) In the paragraph that begins with “D3 procapsids and capsids were found to be larger than those of...” The authors suggest that the removal of the scaffold is what causes the conversion of the prohead conformation to the head conformation. However, the data appears to suggest that the removal is a prerequisite but is not what triggers the conversion. Instead, the data suggests that it is the packaging of the DNA that results in the change. Overall, this section and Fig 2 are very descriptive and doesn’t provide significant insight.

We made no statement, direct or implicit, that removal of the scaffold causes “conversion of the prohead conformation to the head conformation” – which is capsid expansion, i.e., the step from Prohead 2 to Head (or virion). A few lines beyond the quote above, we wrote: “Notably, removal of the scaffold domains correlated with significant changes in the procapsid shell (Fig. 3A,B and Movie S1), illustrating the role of scaffold in overall capsid conformation as described below.” This statement refers to the differences we observe between Prohead 1 (with scaffold) and Prohead 2 (without), a feature that has been rarely described in other phage studies. Later, in the Discussion section “The dynamics of the local 3-fold sites”, we wrote: “Digestion of the scaffold domain triggers conformational differences between Prohead 1 and Prohead 2 that are focused on the local 3-fold sites.” The maturation pathway involves proteolysis (Prohead 1 to Prohead 2) and expansion (Prohead 2 to Head), which we detail in the Introduction and Figure 1A, including this text that describes the trigger for expansion, in agreement with Reviewer #1: “Terminase encapsidates the phage DNA through the portal, triggering expansion of the capsid into the mature form, called Head, where covalent crosslinks form between MCP subunits.” We believe that the points made, especially about changes between Proheads 1 and 2, are novel, significant and insightful, and have been well introduced and are developed in the Discussion.

6) Fig 5 and S6: The text provides a really nice description of the interactions, but some of these are not evident in the figures.

We are glad the Reviewer appreciated our prose but without more explicit guidance we struggle to find fault with these Figures. We have carefully examined each sentence in the paragraph concerned, starting “PDARs from subunits of three different capsomers are organized around each local 3-fold location as trimers (Figs. 5, S6A).”, and find that Figures 5 and S6 well reflect the text, and that attempts at “improving” them lead to complexity that masks comprehension of the interactions. We did elaborate on the nature of the sidechain environment that we had previously limited to “...stabilized by surrounding acidic and polar residues.” as follows:

These two residues are oriented relatively close to one another and may be stabilized by surrounding acidic and polar residues, including E353, E359 & E374 from the I subunit, and E359, D361, D363, F364 and E365 from the F subunit.

We also included some sidechain distance measurements but deferred those to the Results section “Molecular basis of the morphological changes” – see our response to Minor point #17.

7) The authors appear to have interchanged the term P-loop and P-domain. Or P-loop is a region within the P-domain. Either way, P-loop is not defined in the text or figures.

The Reviewer is correct that the P-loop is a region within the P-domain. We have added a label to indicate the P-loop in Figure 1B and defined it as part of the P-domain where it first appears in the text (underlined)

below):

The trimers involving PDARs from hexons, ADC and BHE, include a similar density bridge between two of the subunits (Fig. 5) that correspond to the side chains of R102 (from subunits B and D) and R106 (from subunits E and A) which flank F364 (B/D) from a region of the P-domain called the P-loop (Fig. 5B, top and middle row; Fig. 1B).

8) *“The structural origin of this change is highlighted in Fig. 6B, emphasizing the spine helix (purple), the P-domain (blue) and the PDAR (orange). The P-loop interacts with the PDAR in Prohead 1 but after its removal these P-loops rotate and contact...”*

As depicted in Fig 6 and S7 the described structural changes cannot be assessed by the reader. An overlapping image would help.

While we agree that such a dynamic process is hard to convey with figures, we believe an overlapping image will create more confusion. Thus, we had provided Movie S2 to help illustrate this event (as well as 3 other movies to support other aspects of this work).

9) *“Variations in the angles across Prohead 1 and the changes following proteolysis suggest that the PDAR constraints at the local 3-fold interfaces propagate to the geometry of the shell structure, likely by playing a role in guiding capsid assembly. Together, these observations indicate a local constraining function of the PDAR on the MCP capsid domains. We liken this constraint to a clamp that holds the capsid domain subunits in Prohead 1 to a network of limited interactions and which is released on proteolysis of the scaffold domains.”*

The text beginning at the end of p8 is very speculative.

We have shortened the first sentence quoted above as follows:

Variations in the angles across Prohead 1 and the changes following proteolysis suggest that the scaffold might locally constraint the structure.

A model of how local events might give rise to a global effect remains as in the original Discussion:

“This local rigidity manifests as a distinct angularity of Prohead 1 compared to Prohead 2 (Fig. S9) that is relaxed following removal of the scaffold. Such constraints suggest that the scaffold, and in particular the PDAR, aids in imposing specific dihedral angles between capsomers that effectively govern capsid geometry and size.”

11) S2: *Compare the same regions (the same alpha helix for example) of the four solved structures when demonstrating that the estimated resolution supports the modeling of sidechains.*

Figure S3 (previously S2) now includes modelling of the same alpha helical region – A209 to Y234 – for the four capsid structures.

12) *A local resolution map is needed for all structures. Critical to this is clearly showing the PDAR region given the extensive analysis of this region.*

We have provided a new Supplementary Figure S2 with the local resolution as determined by Resmap. The column of Interior views shows the resolution of the PDAR domains of Prohead 1.

Minor concerns/clarification needed

1) *Why is there so little protease detected on both native agarose and SDS-PAGE gels in fig 2 B, C?*

The inactive protease isn't detected as a separate band on the native agarose gel (Figure 2B) because it is inside the Prohead 1 particles. Although the precise number of protease molecules in Prohead 1 is unknown, measurements from SDS PAGE band intensities suggest ~90 copies (compared to 540 MCPs). The total mass of protease is thus expected to be ~9 times less than the major capsid protein, consistent

with the SDS-PAGE results for D3 proheads in Figure 2C. To support these statements, we included the following in the Introduction:

After completion of Prohead 1, which in the case of HK97 contains 415 copies of the MCP, ~60 copies of the protease [8] and 12 copies of the portal, ...

And added this text to the caption for Figure 2:

Note that the amount of protease was much less than the MCP amount: we calculated that there are ~90 copies of protease per Prohead 1 using band intensity measurements from the P1 gel.

2) *Label capsomer in Fig. 1 as well as define the differences between capsomer, virion, and head. For example, are the virions in Fig. 2E considered capsomers?*

The term “capsomer” is introduced in the second paragraph of the Introduction:

*In the case of the well-studied HK97 coliphage, assembly of the first complete shell, Prohead 1, is believed to be nucleated by the portal, a dodecameric ring around which **hexameric and pentameric capsomers of the MCP** bind with the aid of a 102-residue scaffold domain (also termed the Δ -domain) encoded as an N-terminal extension of the MCP (Fig. 1).*

However, we further highlighted the term by adding a label to Figure 1 that indicates capsomer types.

3) *Fig. 3: provide an overlap of the different hexons (e vs c) to highlight the differences.*

We added a new Supplementary Figure S10 that shows overlaps of the E and C hexon models for each capsid type and refer to it in the Discussion section “The scaffold guides capsid geometry.”

4) *Fig. 4: Panel E is not in the figure legend*

We thank the Reviewer for pointing this out – now corrected. The text already correctly referenced the other panels.

5) *Fig. 9: RMSD*

The Reviewer’s intent is not clear, but we assume it is to calculate RMSDs between the PDARs and/or P domains of HK97 Prohead 1 and D3 Prohead 1. Figure 9 supports discussion on differences in overall organization between the PDAR trimers of HK97 and D3, which are significant given that one has T=7 geometry and the other T=9. We don’t believe that RMSD values contribute usefully to this discussion, especially as they will be large simply due to the different capsid curvatures.

6) *Include the estimated sizes of each domain in panel Fig1A.*

This comment is also unclear – which domains? Of the MCP, protease and portal, or of the proteins themselves? We assume proteins, and this information has now been included in the caption.

7) *Fig1D—include a ladder that was run in parallel with the P1 and P2 fractions. Clearly delineate that the two samples are from cropped gels by separating them and/or placing a dotted line between them.*

We thank the reviewer for this observation. We realize that an over enthusiasm for graphical tidiness caused us to put these two lanes together when they are very clearly not from the same gel. This is now corrected in Figure 2D (not 1D as stated) and we also included the molecular weight ladders from the original gels.

8) *Does the presence or absence of packaged DNA occur randomly in a prep of Virions?*

imaged by cryoEM for structure determination, in conjunction with a purified virion sample that included particles with and without packaged DNA (Fig. 2E).

This comment seems to be missing some text, but the first line is clear – some virions expel their DNA during purification probably due to the presence of bacterial debris. The remainder is a partial quote from the Results “D3 capsid morphologies” section, but the Reviewer’s intent is unknown.

9) *Recommend not using "Head" as the description of the empty head of the virion. Empty-Head virion would be clearer to the naive reader.*

We agree and had used “empty Head” throughout, with two exceptions (Table 1, Figure S3 caption) – now corrected.

10) *Please provide a scale bar on the orthoslice images in Fig 2.*

The Reviewer means Figure 3, and we have added back the scalebar that was inadvertently omitted.

11) *Provide the measured diameter of each and the observed change. “small but distinct increase of shell diameter”*

These are already shown in Supplementary Figure S11A (previously S3A), and several measurements are given as the changes are not uniform across the capsids.

12) *What is meant by “most pronounced at the icosahedral faces”? Is the diameter the same at the 3-fold interface between head and virion? Again, scale bars and overlaid measurements would provide the reader with the needed information.*

As for point 11 above, measurements are already given in Supplementary Figure S11.

13) *Figure 3: indicating where the symmetry axis described in the text are on the map would be helpful.*

Symmetry axes were already indicated on Figure 3A using appropriately symmetric yellow shapes: pentagon, 5-fold axis; triangle, 3-fold axis; and lens, 2-fold axis. We have now replaced these shapes with the numbers 5, 3 and 2, respectively, for clarity.

14) *Figure 3C,D are not discussed in the text in a meaningful way and so are they needed?*

Figure 3C defines the asymmetric unit and the subunit labeling scheme (A-I) and so is essential to orient the reader. Figure 3D is the sole demonstration in the main manuscript of the quality of density and accuracy of the modeling, aside from Fig 4D and 5B, which display only the density for the PDAR sub-domain. All representations of map and model quality are otherwise consigned to Supplementary Figures S1 and S3. We believe some evidence of this is useful without recourse to the Supplementary Data and we propose to maintain both panels unless overruled by an Editorial decision.

15) *PDBs and Maps were not available to download. (may be an issue with the portal).*

Density maps and models were deposited in the EMDB and PDB repositories, respectively, as required by Nature Portfolio policies, and accession IDs are listed in Table 1. We have added a new section “Data Availability” that indicates this. However, we did not previously upload data to the figshare repository for review but have now done so with this resubmission. As noted in response to Reviewer #2’s point 1, we have also included the PDB validation reports although they are clearly watermarked with the statement: “This wwPDB validation report is NOT for manuscript review.”

16) Fig 7: The viewing angles chosen to depict the organization of R102 and F364 in prohead 1 does not clearly show the distance or relationship between the two amino acids clearly. This is not an issue for the viewing angles used for prohead 2.

The viewing angles are meant to be approximately equivalent, but we acknowledge that representing a three-dimensional organization in a Figure is challenging. However, Movie S2 was already provided to compensate for limitations of the 2D representation.

17) A zoom in and distance measurement of the change of this described change would be good. “After scaffold removal abolishes any R102/F364 interaction, F364 is now adjacent to another arginine, R372,…”

In place of a zoomed-in inset that would further complicate Figure 7, we had referenced Movie S2 at the start of the paragraph, which clearly shows this change. However, we have now added the following measurements to the text quoted above, as follows:

In Prohead 1, R102 of the PDAR is situated close to F364 of the same subunit, with sidechain distances ranging from 3.3-4.3 Å and possibly forming a cation-Pi interaction [23]. After scaffold removal abolishes any R102/F364 interaction, the F364 sidechains are now located within 3.3-4.6 Å from those of another arginine, R372, that was previously distant, as well as to N360, both of an adjacent subunit.

18) Fig3: Indicate the final determined resolution for each map. The dotted line denoting the different maps shown in B does not line up with the seam of the pasted images.

The resolution and statistics were already given in Table 1. The differences across the dotted lines reflect changes in the capsids from Prohead 1 to 2 (proteolysis) and virion to Empty Head (loss of DNA). The two halves are simply not identical in each case.

19) Fig7: Clearly label the monomers for the alphabet three folds.

We added a label in the center of each triad that indicates the subunits.

20) S6: the orientation of “CDA” in panel-A does not match “CDA” in panel-D

The Reviewer is quite right – the triad is rotated ~120-degrees and we have corrected panel D.

21) S3: were the measurements from the backbone or side chains, and if side chains are the modeled positions supported by density in the map.

The measurements are from the backbone, which seems the most appropriate reference point, although the sidechains are often clearly visible.

22) S7: “Surfaces are colored by radius as shown in the scales” radius relative to what?

The accepted understanding of “radius” in the context of a spherical virus capsid would be from the center of the capsid. Nonetheless, we have added that clarification to the caption.

23) Fig. 6: Please clarify with a cartoon the difference between up and down conformations. Or place an arrow pointing which feature is up/down.

We appreciate the importance of this point. It is clarified in Supplementary Figure S7 (red dots), and we added an inset in Figure 6 to show the difference, as suggested.

Reviewer #2 (Remarks to the Author):

This manuscript describes four cryo-EM structures capturing key assembly intermediates of the Pseudomonas phage D3 capsid: the initial Prohead 1 (with scaffold), the proteolyzed Prohead 2 (scaffold removed), the DNA-filled mature virion, and the empty head. The MCP's N-terminal scaffold domain serves as a "molecular clamp" on the interior capsid surface, whose constraint on subunit interactions in Prohead 1 is critical for determining the final T=9 capsid size. Subsequent proteolytic cleavage of the scaffold releases this "molecular clamp," triggering the formation of new, more stable inter-subunit contacts that prime the capsid for DNA packaging and expansion.

The authors provide high-quality structural data and a detailed description of the phage D3 assembly process. However, the work requires major revisions before it can be considered for publication. There are concerns regarding its data validation, method, and discussion. The major issues are as follows:

1. The structural biology research requires the provision of sufficient validation data, which is currently lacking in this manuscript.

Full PDB validation reports are not provided. While MolProbity scores are provided in Table 1, the full PDB validation reports are not included, making it impossible to assess the detailed quality of the atomic models in terms of bond lengths, bond angles, Ramachandran plots, and clashes. Full validation reports should be provided as supplementary material upon submission.

Local Resolution Maps are absent. The manuscript only reports the overall resolution, which can mask variations in quality across different regions of the structure. Without them, it is impossible to judge the quality of the density in key areas, such as the flexible E-loop or the functionally critical PDAR and P-loop interfaces.

We are happy to supply the full PDB validation reports. We did not so initially as they are clearly watermarked with the statement: "This wwPDB validation report is NOT for manuscript review". We don't know why they are not intended for review.

Regarding local resolution maps – please see our response to Reviewer #1, minor point 12: We have provided a new Supplementary Figure S2 with the local resolution as determined by Resmap.

2. The description in the Methods section is too brief and lacks critical details.

"Cryo-electron microscopy" Section: The description is too general. The authors should provide more detailed parameters, such as the specific defocus range used during data collection; the specific methods and software parameters for particle picking; the strategies and rounds for 2D and 3D classification; and the parameter settings for final reconstruction and refinement (Refinement/Polishing) in RELION.

"Structural analysis" Section: This description is also unclear. The authors mention using ModelAngelo and ISOLDE, but these are just tool names. The workflow for model building and refinement using ModelAngelo and ISOLDE needs to be described in detail, including how flexible regions were handled.

Reviewer #1 asks for similar detail in their Intermediate concern #4 – a pipeline figure for cryoEM analyses – please see our response there. Regarding the structural analysis, phage capsids are very straightforward to analyze compared to small complexes or membrane proteins – they are large and obvious for simple automated picking; classification is simple in 2D; initial models are also derived with ease; and the particles tend to be uniform leading to good resolution. Thus, there is little of interest to report here beyond very standard processing, as already mentioned in the Methods. We added a brief description of our focused alignment by re-picking the region around the 12 vertices of each capsid, but the methodology is covered in our previous work on phage HK97, as cited, and doesn't need repeating in

full here. We also expanded our description of model building and refinement with ModelAngelo and ISOLDE.

3. The authors mention performing local focused refinement on the five-fold vertices, which improved the resolution in these areas. However, the uniqueness of the $T=9$ lattice lies precisely in the two non-equivalent hexamers: the edge hexon (eH) and the central hexon (cH). These hexons are located at or near the two-fold and three-fold axes. The authors do not mention whether they also performed local focused refinement on these equally critical regions. If optimization was only performed on the five-fold axes, it could lead to biases in the interpretation of the conformations and interactions of the hexameric regions (especially the much-discussed cH), and their model reliability would be lower than that of the five-fold regions. The authors should clarify their refinement strategy for the different symmetry axes and the final local structural quality.

As mentioned in the text, and now in more detail in the updated Methods section, we calculated new vertex-based maps after completing icosahedral refinement of the entire capsids. By extracting twelve smaller vertex-based regions and reconstructing these with C5 symmetry imposed (but 12-fold more “particles”) we reduce the distortion of focus gradients through the larger capsid particles and can filter out vertices that may be of lower uniformity. However, we did not perform a focused refinement centered on the cH (3 fold-axis) since the resolution of the ASU density – which includes two cH subunits – is sufficient to build a reliable ASU model. The local resolution estimates in the new Supplementary Figure S2 show that the quality of the density extends throughout the asymmetric unit.

4. The manuscript contains several instances of incomplete or erroneous literature citations. For example, on page 6, “Before capsid expansion, eHs adopt a pseudo-2-fold symmetry (Fig. 3B) typical of the skewed conformation observed for the single hexon type in $T=7$ proheads 5,19”: The citation here is problematic, being primarily outdated and not precise enough. The cited reference 19 is an early, low-resolution study. Subsequent work (e.g., studies on the capsids of phages HK97, P22, lambda) has revealed the details of the “skewed hexon” conformation at high resolution. The authors should update this citation to include more recent, high-resolution representative studies.

Another example on page 11: “Furthermore, capsid size correlates with the length of the coiled-coil subdomain, as exemplified by phages HK97 ($T=7$), D3 ($T=9$) and T5 ($T=13$).” This statement lacks any citation.

On page 12: “Expansion results in further conformational changes at the local 3-fold sites, as previously observed in other dsDNA phage capsids such as HK97, T5, λ , P22 and T7.” This sentence is also missing references.

On page 13: The citation 37 in “...strengthen the D3 capsid in a manner similar to the 3-fold binding decoration proteins of non-crosslinking phages such as λ .37” is incorrect. A more appropriate reference matching the description would be (Structure. 2008, 16(9):1399-406 or Nat Struct Biol. 2000, 7(3):230-7).

We will happily update the references but also point out that – for example – the early work on lambda [19, now 20] and HK97 [5] clearly showed skewed hexamers despite the low resolution, which was a totally unexpected observation at that time. The fact of the skewing has survived 30 years of subsequent study, and those early papers are no less correct today than they were at the time. It is easy to lose sight of the deep history behind these discoveries by referencing only the latest literature. At the point where we cite these two papers, only the fact of the skewing is relevant, not the details. Subsequent text refers to later studies when the details become important to the argument.

The correlation between the length of the coiled-coil subdomain and capsid size is illustrated in Supplementary Figure S4, and the passage quoted by the Reviewer, in full, references that Figure: “Furthermore, capsid size correlates with the length of the coiled-coil subdomain, as exemplified by

phages HK97 (T=7), D3 (T=9) and T5 (T=13) (Fig. S4A)". We are not aware of any study that specifically examines this correlation.

The sentence quoted from page 12 now includes citations as follows:

HK97 (Gertsman et al., 2009)

lambda (Wang et al., 2022)

T5 (Huet et al., 2019)

P22 (Xiao et al., 2023)

T7 (Guo et al., 2014)

We struggle to understand the objection to citation [37, now 43] which documents the role of the gpD protein – the title is literally just that: “Packaging of coliphage lambda DNA. II. The role of the gene D protein”. There the authors state: “pD appears to act by stabilizing the head against disruption by overfilling with DNA rather than by changing the capacity of the head for DNA.” The first paper suggested as a substitute, Lander et al 2008, does not measure stability of the lambda capsid in the presence or absence of gpD, but it does suggest how it might effect stabilization – we have now included it as reference 44. We are, of course, familiar with the second paper, Yang et al, 2000, on which corresponding author Conway is also an author. However, this work was based on a crystal structure of the gpD trimer and does not address stabilization or details of its interactions with the capsid.

5. The authors note that while inclusion of the portal gene enhanced particle yield, the portal protein itself was not detected in the purified procapsids. Under physiological conditions, the portal is the initiation point and nucleation center for capsid assembly (Sci Adv. 2023, 9(24):eadg8868; PLoS Biol. 2025, 23(4):e3003104). Its 12-fold symmetry and the inherent mismatch with the 5-fold axes of the capsid has effects on the entire geometric pathway of assembly. The T=9 capsids observed in this study are the products of spontaneous self-assembly via overexpression of high-concentration protein, without portal nucleation. Therefore, it cannot be ruled out that this in vitro self-assembly pathway and its final product are non-physiological artifacts of the experimental system. The authors should address this limitation in the discussion and cautiously argue the extent to which their observed structures represent the true in vivo assembly process.

Whether the portal is the nucleation center for capsid assembly *in vivo* is still very much a matter of debate. The Reviewer cites our 2023 paper on the HK97 Prohead 1 portal structure (Huet *et al*, Sci Adv. 2023 – [11]) in support of this function, but we do not establish it there, and our Introduction here states: “In the case of the well-studied HK97 coliphage, assembly of the first complete shell, Prohead 1, is believed to be nucleated by the portal”. Indeed, a nucleating portal is clearly not required for the D3 Prohead 1 to assemble in our experimental conditions or to progress through proteolysis to Prohead 2, and this is also true for HK97 (see below). This raises a more interesting question – how is the portal so efficiently incorporated *in vivo* when the major capsid protein can efficiently assemble without it? We speculated in our 2023 HK97 paper that the portal might accelerate formation of the capsid around it, thus favoring its incorporation by modifying the assembly kinetics rather than by effecting a change in the assembling particles. We are exploring this hypothesis, but it is not a subject for the current manuscript.

Regarding the authenticity of our portal-free procapsids, we first note that the assembly is done in *E. coli*, not *in vitro*, and that previous experience with portal-free HK97 MCP is that the Prohead 1 particles have the correct size and shape, and are competent to proteolyze to Prohead 2 and expand to the mature Head. In addition, we note that two recent HK97 studies done by us on the portal of Prohead 1 (Huet et al, Sci Adv 2023 [11]) and with collaborators on Prohead 2 with and without terminase (Hawkins et al Nuc Acids Res 2023) have shown that the previously established structures of the proheads with and without portal are the same when icosahedral symmetry is imposed, confirming the validity of the previous work. Indeed, the assembly pathways for T=7 and T=9 capsids are likely to have differences according to the geometries, but the question we address in this work is not about the pathway but about capsid size

regulation and the similarities and differences with our new D3 structures and those of the related but smaller HK97. We thus believe that both the Prohead 1 and the Prohead 2 particles we obtain by *in vivo* plasmid overexpression are relevant and representative of the equivalent particles produced during the phage infection, just minus the portal but otherwise exhibiting the expected size and well known prohead features, as described in the text. We have modified the first paragraph of the Discussion to cover some of these points:

Assembly of capsids with the authentic size and shape requires only the full-length MCP that includes the N-terminal scaffold and C-terminal capsid domains, and as for phage HK97 the portal is not required. It should be noted that the HK97 assembly pathway was established without the portal and only recently has this special vertex been included in assembly analysis studies [11, 28]. In particular, structures of capsids with or without the portal were virtually identical when icosahedral symmetry is imposed, demonstrating the relevance of portal-less capsids for studying assembly.

6. Some structural descriptions appear to be based on uncertain observations. For example, the claim that an "isolated region of density" corresponds to the N-arm in Prohead 2 is difficult to assess, as it could not be modeled due to insufficient resolution (Figure S8).

The speculation that a "cation-pi interaction may form" between R102 and F364 in Prohead 1 is stated without providing the necessary data (distances) for validation. The authors should provide these details or show the fit of the side chains into the density.

The entirety of this passage connects this density to equivalent density on the HK97 capsid: "While this new density was insufficiently resolved to assign sidechains, its location adjacent to the C-terminus of the scaffold in Prohead 1 suggests that it corresponds to the MCP N-arm, as has been observed in HK97 [11, 22]." While speculative, it is not without a good basis and we would be remiss not to make this suggestion by ignoring it.

Regarding the "cation-pi interaction", please see our response to Reviewer #1's minor comment #17 where we add new text listing distances between these sidechains.

7. In the discussion, the authors attempt to generalize their findings from D3 and HK97 (both with N-terminally fused scaffold proteins) to all dsDNA phages, including those with separately encoded scaffold proteins. There are differences in stoichiometry and conformational constraints between covalently linked and independently expressed scaffold proteins (e.g., covalent linkage ensures a 1:1 ratio and a fixed topological position). The authors should not simply equate the two but should discuss the differences between the systems in more detail and carefully define the scope of their conclusions.

The Reviewer refers to our Discussion section "Generalization". However, we are not certain how this text is lacking in the Reviewer's eyes. We make specific connections to structures of phages with separately encoded scaffold, which are very few due to the difficulty of such studies, but including the T=7 phages 80alpha and P22. We detail how mutants or viral "pirates" modify the capsid size in these systems through modifications to contacts that correspond to PDAR-capsid sites of interaction. Indeed, our Discussion on this point is strictly limited to what homologies we can observe between the D3/HK97 data and the 80alpha/P22 data (with some less relevant information from the small prolate phi29 phage) – there really exists no other data to compare. We derive cautious conclusions from this and propose additional studies. We don't see what changes would improve the text.

Overall, the manuscript describes the structure of a T=9 phage capsid through its assembly pathway at near-atomic resolution. The work provides a solid structural basis for understanding how the scaffold domain acts as a "clamp" to guide capsid maturation, largely confirming the operation of known assembly mechanisms in a new system. This could be an interesting work, but considerable revision is required to address issues of data validation, methodological detail, and to more clearly define its specific contribution to the problem of capsid size determination.

We appreciate the Reviewer's interest but suggest that the points raised are of much lesser concern than articulated here, as we have described in our previous responses. We strongly disagree that our work is largely confirming a known mechanism – this is simply incorrect as there are few phages where a Prohead 1 particle can be distinguished experimentally from a proteolysed Prohead 2 particle for making the evaluations we present here. Indeed, the HK97 paradigm where the scaffold is fused to the MCP is particularly advantageous for this work, as we state, and phage D3 is only the second where this has been studied after our own work on HK97 [11 – Huet et al, 2023]. Indeed, the D3 analysis has added considerable perspective to our existing HK97 models and we can now draw conclusions that were not evident previously.

REVIEWER COMMENTS

Reviewer #2 (Remarks to the Author):

The authors have addressed most of the concerns raised and implemented necessary revisions. I have no further comments.

We thank the reviewer for their detailed comments.

Reviewer #3 (Remarks to the Author)

Review of revised manuscript by Belford et al. NCOMMS-

This manuscript describes the structures of the prohead and mature head of HK97-like Pseudomonas phage D3. The most interesting aspect of the work is their identification of the scaffolding protein and in particular the so-called PDAR (an acronym I am not fond of).

Reviewer 1 had a rather long list of comments and concerns. The main concern regarded the lack of non-structural experimental validation of the authors' hypothesis of the involvement of the PDAR in capsid maturation, as well as numerous problems with the presentation and interpretation.

In my opinion, the authors have addressed the reviewer's concerns adequately in the revised manuscript and in their rebuttal. Specific concerns and the authors' response include:

1. The reviewer proposed experimental verification of the importance of residues R102 and R106 in the PDAR by mutational analysis. The authors' response was that this is outside the scope of the current structural paper. I tend to agree with the authors in this point.

2. "Provide images... of the map/model ... of the 3fold interface..." The authors have provided such images in Figure 5B, which show clearly the details of the 3fold interfaces.

3. Reviewer 1 had concerns about the interpretation of the protein expression gels in Fig 2. The authors explain that the figure is only meant to show the expression of the various proteins from the plasmid, not

for quantitation of expression. I think the experiment shown is fine, since the bands on the gel are pretty unambiguous and the main point is the presence of the proteins in the reconstructions. Arguably, the proteins present in the D3 virions should have been identified by MS analysis, but this is not a major concern, since the focus in the paper is only on the major capsid protein.

4. "No attempt to model the scaffolding region was made." While modeling was not feasible, the authors have included an additional AlphaFold prediction of the scaffolding that matches the (unresolved) part of the towers seen in the map. Text has been added to explain the approach.

Other minor/intermediate concerns were addressed adequately. I have no particular issues with the revised manuscript.

We appreciate the reviewer stepping in at this late stage and reviewing the revised manuscript as well as the comments by the absent Reviewer #1. Since there is no specific point that requires a response, we have none.